# Grounding Neural Inference with Satisfiability Modulo Theories

**Zifan Wang**[*†]
Center for AI Safety
zifan@safe.ai

**Saranya Vijayakumar**[*]
Carnegie Mellon University
saranyav@andrew.cmu.edu

**Kaiji Lu**[†]
Pinterest Inc.
Caleblu95@gmail.com

**Vijay Ganesh**
Georgia Institute of Technology
vganesh@gatech.edu

**Somesh Jha**
University of Wisconsin-Madison
jha@cs.wisc.edu

**Matt Fredriskon**[‡]
Carnegie Mellon University
mfredrik@cmu.edu

## Abstract

Recent techniques that integrate *solver layers* into Deep Neural Networks (DNNs) have shown promise in bridging a long-standing gap between inductive learning and symbolic reasoning techniques. In this paper we present a set of techniques for integrating *Satisfiability Modulo Theories* (SMT) solvers into the forward and backward passes of a deep network layer, called SMTLayer. Using this approach, one can encode rich domain knowledge into the network in the form of mathematical formulas. In the forward pass, the solver uses symbols produced by prior layers, along with these formulas, to construct inferences; in the backward pass, the solver informs updates to the network, driving it towards representations that are compatible with the solver's theory. Notably, the solver need not be differentiable. We implement SMTLayer as a Pytorch module, and our empirical results show that it leads to models that *1)* require fewer training samples than conventional models, *2)* that have consistent performance against certain types of covariate shift, and *3)* that ultimately learn representations that are consistent with symbolic knowledge, and thus naturally interpretable. Our code is available at https://github.com/cmu-transparency/smt-layer

## 1 Introduction

Deep neural networks (DNNs) have recently made significant strides, achieving surprising levels of performance on tasks like question-answering [40], text summarization [27], and code generation [4, 21]. However, the ability of models that perform well on these benchmarks to consistently demonstrate sound logical reasoning, even on tasks that may appear to be more simple and self-contained, remains in question [6, 31, 39]. For example, a prediction market on whether GPT-4 will be able to consistently solve "easy" Sudoku puzzles from the LA Times has remained open for several months at the time of this writing, despite the prompt-tuning efforts of traders on the market [19].

One way to address this issue is to encode logical constraints that are essential for certain inference tasks symbolically, making them available to the model either during training or inference. A promising approach to this type of "neuro-symbolic" learning is to integrate *solver layers* into the model [35, 34, 13, 18], both during training and inference. During training, the model must learn a representation of the data that is compatible with the symbolic knowledge, and during inference this

---

[*]Equal contribution.

[†]Work done as a PhD student in Carnegie Mellon University.

[‡]Corresponding author.

37th Conference on Neural Information Processing Systems (NeurIPS 2023).

*grounded* representation is used to provide information from which a symbolic reasoning engine (i.e., a "solver") can extract accurate results.

The most straightforward way to incorporate a solver layer into a deep model is to learn representations that are compatible with symbols used by the solver. For example, if one wanted to leverage symbolic domain knowledge to classify images of birds, or diagnose ailments from CT scans, then one could train a model as in "concept bottlenecking" [14]. This requires detailed supervised labels, which may be prohibitively expensive to obtain and keep consistent with a potentially evolving domain theory.

We present a set of techniques for incorporating a *Satisfiability Modulo Theories* (SMT) solver into a DNN layer so that symbolic knowledge can be leveraged to learn such a compatible representation, *without requiring label supervision*. Our method, unlike prior methods, does not require fine-grained labels to learn representations [14]. For example, for learning Visual Sudoku, regular supervised learning requires labels of individual digits, rather than just the solution to a set of Sudoku puzzles, to learn a dedicated digit classifier first. Our work sidesteps the need to manually break a problem apart and obtain intermediate labels to supervise on, and instead allows for learning a predictor end-to-end from just labels on the targeted task. Our approach is general, and can handle a broad range of domain knowledge encoded as SMT constraints, provided that they interface with the surrounding neural network layers over propositional variables. Unlike related prior work [35], our approach does not approximate the solver's behavior by formulating a differentiable relaxation. Rather, as the solver works on a set of constraints, we extract information that is geared towards checking the correctness of preceding layers, and use that information to construct training updates (Section 4.2).

We present two different approaches for this, one based on unsatisfiable cores, and another based on weighted MaxSMT (Section 4.2). There are several advantages to this approach. Aside from the mild interface constraints mentioned earlier (i.e., solver and neural layers interface with each other via boolean variables), our approach does not place any restrictions on the theory solver embedded in the layer, such as linearity [34] or even decidability—if the solver is capable of efficiently discharging the relevant constraints, then the layer can operate as intended. Because there is no need to provide a differentiable relaxation for each theory or solver technique that one may want to incorporate, we can leverage the continuous and unabated progress being made in solver technology.

We implement our approach as a PyTorch [22] layer, using the Z3 [7] SMT solver as the solver layer to solve SMT and MaxSMT constraints. On three applications involving vision and natural language: visual arithmetic, algebraic equation solving, and a so-called natural language "liar's puzzle," we demonstrate that our implementation can be incorporated into DNN architectures to solve problems more effectively than conventional DNNs (Section 5). In particular, our results show that the data needed to train a DNN with symbolic knowledge may be much simpler than may be necessary otherwise, and that while doing so is more expensive computationally, often times the more efficient (i.e., not involving MaxSMT) algorithms perform well in practice.

Our contributions are as follows:

1. We present SMTLayer, a framework for incorporating an SMT solver into a DNN, as a layer that leverages symbolic knowledge during training and inference.

2. Our empirical evaluation, over four types of tasks requiring logical reasoning, e.g. visual Sudoku, shows that models using SMTLayer require significantly less training data, can be trained more efficiently, and generalize a lot better on the much larger test sets compared to those based on closely-related prior work [35, 13].

Section 3 provides background, Section 4 describes SMTLayer, Section 5 gives our empirical evaluation, and Appendix A.1 (supplementary material) discusses the broader impacts of our work.

## 2 Related Work

**Solver Layers.** Vlastelica et al. [34] integrate a black-box and non-differentiable combinatorial solver on top of a deep network. To propagate the gradient through the solver on the backward pass, they linearly interpolate the loss w.r.t the solver's input and define the gradient of the solver as the slopes of the line segments. CSL solves a set of problems where the solver's objective must be linear w.r.t its input, e.g. finding the shortest path and travel salesman problem (TSP). Further, the authors assume that the only labels available are the outputs of solvers, e.g. the minimum cost in TSP, and

hence their tool has to discover the label for the output of the network itself. These requirements limit the choices one has for the solver layer. Wang et al. [35] present SATNet, a network architecture with a differentiable approximate MAXSAT solver layer. Their approximation is based on a coordinate descent approach to solving the semidefinite program (SDP) relaxation of the MAXSAT problem. SATNet does not assume that the logical structure of the problem is given, and instead attempts to learn it.

**Neural Logic Programming.** While SATNet integrates a logic-based solver on top of a network, DeepProbLog takes the opposite approach, extending the capability of a probabilistic logic programming language with neural predicates [18]. In the context of our work, the logic program can be viewed as a "solver layer" that explicitly encodes symbolic knowledge. Scallop [13] extends DeepProbLog to scale without sacrificing accuracy compared to DeepProbLog. Similarly to Deep-ProbLog, each possible result of the sum of two digits in MNIST is given a probability, in the form of a weighted Boolean formula. They prune unlikely clauses of the formula, represented by proofs, only keeping the top-$k$ most likely. Likelihood is computed using weighted model counting [13, 3]. These techniques are well-suited to problems that benefit from probabilistic Datalog, but have inherent limitations: they cannot handle quantifiers, general negation, and the range of supported first-order theories is more restrictive.

More recent work explores different directions in training models to perform neural-symbolic reasoning. For example, SMT solvers and MCMC sampling are utilized to support network training, side-stepping shortcut satisfaction [15, 16]. Hoernle et al. [12] introduce MultiplexNet, a deep model that ensures the satisfaction of symbolic constraints during the inference stage and adds an additional dimension to the neuro-symbolic learning landscape. Marconato et al. [20] has investigated how to mitigate the challenges and limitations associated with neuro-symbolic reasoning shortcuts and Yang et al. [37] use straight-through-estimators and logical constraints for neural network learning.

**Differentiable Logic & Semantic Losses.** Another recent direction has explored differentiable logics [9, 33, 32], loss functions that incorporate logical semantics [1, 36], or training methods guided by symbolic solvers [10, 28]. These approaches provide ways of integrating symbolic knowledge into training, by making logical formulas differentiable, and therefore amenable to optimization when included in a loss function. This line of work does not explicitly aim to make use of symbolic information during inference. In contrast, the information that our approach extracts from the solver during training is used to condition the model towards a representation that will allow it to communicate effectively with the solver during inference. Additionally, we do not require the logical formulas, or the solver, to be differentiable.

## 3 Background

Let $\mathbf{X}$ denote a domain of features, $\mathbf{Y}$ a domain of labels, and $\mathcal{D}$ a distribution over $\mathbf{X} \times \mathbf{Y}$. Formally, $\mathcal{D}$ is a probability measure on a space given by a $\sigma$-algebra over subsets of $\wp(\mathbf{X} \times \mathbf{Y})$. The goal of a learning algorithm $A$ is to find a function $h : \mathbf{X} \to \mathbf{Y}$ that, for $(x, y) \sim \mathcal{D}$, can be used to predict $y$ when given $x$. To do this, $A$ is given a set of training examples $S = (x_1, y_1), \ldots, (x_m, y_m)$ sampled i.i.d. from $\mathcal{D}$, and uses some criterion to select $h$ from a hypothesis class $\mathcal{H}$ of functions. We refer to $h$ as the *model* learned by $A$ on $S$. When the learning algorithm $A$ is clear from the context, we will write $h_S$ to denote the model produced from the given sample. Throughout this paper, we will generally assume that the loss is either the 0-1 loss $\ell^{01}$ or binary cross-entropy $\ell^{\mathrm{bce}}$.

A theory $T$ consists of a signature $\Sigma$ of constant, predicate, and function symbols, as well as a set of axioms over $\Sigma$. Formulas in a theory are composed of elements of $\Sigma$, variables, and logical symbols such as quantifiers and Boolean operations. We use the term *decision procedure* to refer to an algorithm that is given an open $T$-formula, and returns *true* if it is satisfiable, and *false* otherwise. Additionally, it may return an assignment to all of the variables that demonstrates satisfiability, or if the formula is not satisfiable, then it may return an *unsatisfiable core*, which is a subset of clauses taken from the formula's representation in conjunctive normal form that remains unsatisfiable. Loosely, we also refer to such an algorithm as a "solver", but this term is more general, and could also refer to an algorithm that identifies the maximal set of clauses, possibly weighted by some user-defined values, that are satisfiable when conjoined.

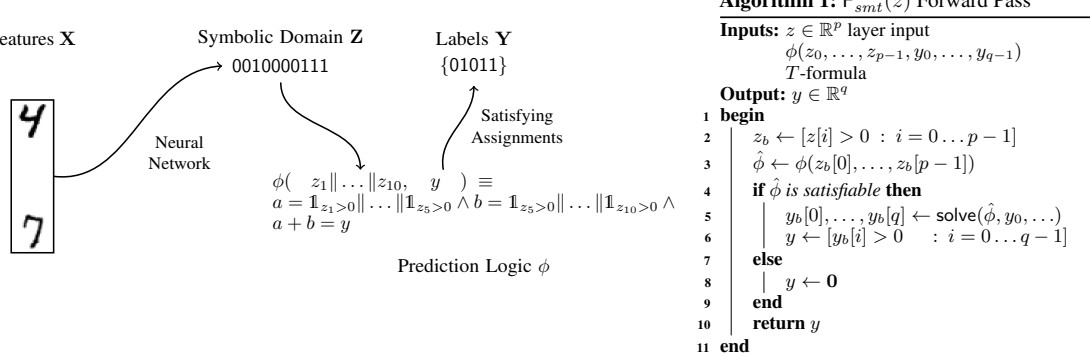

**Algorithm 1:** $\mathsf{F}^{\phi}_{smt}(z)$ Forward Pass

**Inputs:** $z \in \mathbb{R}^p$ layer input
$\qquad\quad \phi(z_0, \ldots, z_{p-1}, y_0, \ldots, y_{q-1})$
$\qquad\quad T$-formula
**Output:** $y \in \mathbb{R}^q$

```
1  begin
2  │   z_b ← [z[i] > 0  :  i = 0 … p − 1]
3  │   φ̂ ← φ(z_b[0], …, z_b[p − 1])
4  │   if φ̂ is satisfiable then
5  │   │   y_b[0], …, y_b[q] ← solve(φ̂, y_0, …)
6  │   │   y ← [y_b[i] > 0    : i = 0 … q − 1]
7  │   else
8  │   │   y ← 0
9  │   end
10 │   return y
11 end
```

Figure 1: Illustrative MNIST addition example (left), and SMT-based forward pass (right, Alg. 1). We use a binary encoding of digits, because SMT solvers support constraints involving integers that are encoded this way. While only four bits per digit are necessary to represent the inputs, five are needed for the output, and we represent all digits as 5-bit vectors for uniformity.

## 4 Constructing SMTLayer

In this section, we present SMTLayer, a set of algorithms for computing the forward and backward passes of a layer whose behavior is defined by a set of user-defined SMT constraints. SMTLayer does not have trainable parameters, and its functionality is wholly defined by a set of SMT constraints $\phi$ that are provided by the model designer. SMTLayer can be used in modern deep-learning frameworks as a drop-in replacement for more conventional neural network layers, e.g., dense, convolutional, and LSTM [11] are prominent examples of widely-used layers.

Section 4.1 provides a high-level overview of our approach, Section 4.2 describes them in detail, and Section 4.3 begins an analysis of this setting that we hope future work will continue developing.

### 4.1 Overview

We envision SMTLayer being used primarily at the top of a DNN taking inputs from a stack of conventional DNN layers that convert raw input features into ground terms for the constraints $\phi(z_0, \ldots, z_{p-1}, y_0, \ldots, y_{q-1})$ embedded in SMTLayer, and producing outputs that are consistent with $\phi$ and the given ground terms. Figure 1 shows an illustrative example, with the previously-studied problem of MNIST addition [18, 13].

During the forward pass the outputs of the previous layer are mapped to designated free variables $z_0, \ldots, z_{p-1}$. The layer then checks the satisfiability of $\phi$, a formula in an appropriate combination of first-order theories, after substituting these ground terms for the $z_i$, and the output of the layer consists of the solver's model for $y_0, \ldots, y_{q-1}$. These outputs are converted from Boolean to floating-point values by mapping *false* to -1 and *true* to 1. At the moment, the only restriction on $\phi$ that our layer requires is that $z$ and $y$ be vectors of Booleans, so that they can be appropriately mapped to continuous values; any other symbols appearing in $\phi$ can come from arbitrary domains (e.g. strings) supported by the underlying SMT solver.

In the backward pass, the layer receives the gradient of its output with respect to the function whose derivative is being computed, which we will assume is the binary cross-entropy loss $\ell(y, y^\star)$. Unless stated otherwise, we will assume this loss for the remainder of the section. This gradient is used, along with the inputs and outputs of the corresponding forward pass, to first compute an amended output $\hat{y}$ which corresponds to an output that would have yielded a smaller loss. Because the outputs are Boolean, it is always possible to determine the ground truth output $y^\star$ from this information. Using $\hat{y}$, the layer determines which of components of its inputs are inconsistent with $\phi$ and $\hat{y}$, and provides the corresponding gradients to the previous layer. Section 4.2 details the manner in which these gradients are computed.

## 4.2 SMTLayer, forward and backward

We now present the details of the forward and backward passes of SMTLayer. There are two algorithms for each pass, $\mathsf{F}_{smt}^{\phi}$ and $\mathsf{F}_{max}^{\phi}$ are forward passes, and $\mathsf{B}_{core}^{\phi}$, $\mathsf{B}_{max}^{\phi}$ are backward passes. $\mathsf{F}_{max}^{\phi}$ and $\mathsf{B}_{max}^{\phi}$ both make use of MaxSMT solvers, whereas $\mathsf{F}_{smt}^{\phi}$ and $\mathsf{B}_{core}^{\phi}$ rely on satisfiability solvers (SMT). They are all compatible with each other. That is, $\mathsf{F}_{smt}^{\phi}$ (and $\mathsf{F}_{max}^{\phi}$) can be used with either $\mathsf{B}_{max}^{\phi}$ or $\mathsf{B}_{core}^{\phi}$. Algorithms for $\mathsf{F}_{smt}^{\phi}$ and $\mathsf{B}_{core}^{\phi}$ are included in this section, and the MaxSMT-based variants are detailed in Appendix A.2.

**Forward pass.** Algorithm 1 illustrates $\mathsf{F}_{smt}^{\phi}$, our SMT-based forward pass, and Algorithm 3 for computing $\mathsf{F}_{max}^{\phi}$ is given in Appendix A.2. Both of the algorithms are parameterized by a user-provided first-order formula $\phi$, and take a single vector-valued input consisting of unscaled floating-point values (*logits*). These values are cast to Booleans by taking their sign on line 2 of both algorithms, and they are equated with the corresponding free variables $z_0, \ldots, z_{p-1}$.

The key difference between $\mathsf{F}_{max}^{\phi}$ and $\mathsf{F}_{smt}^{\phi}$ is the way in which they handle inputs that are inconsistent with $\phi$ when interpreted as Booleans. $\mathsf{F}_{smt}^{\phi}$ addresses this by providing an output that is also inconsistent with $\phi$, i.e. a vector of zeroes, effectively signaling that the network below it did not provide consistent inputs. Alternatively, we can interpret the values provided by the network as Booleans enriched with "confidence" values. To obtain MaxSMT weights, Algorithm 3 scales the input logits to a formal probability distribution via the softmax function (line 3). With this approach, SMTLayer will always provide a valid, although not necessarily correct, output that is consistent wrt $\phi$ with the inputs of which the network below is most "confident." (line 4).

**Backward pass.** The backward pass is responsible for computing the gradient of the loss with respect to the layer inputs. It is given the gradient of the loss with respect to the layer outputs, and is assumed to have memoized the inputs and outputs from the forward pass. The gradients returned by this pass are then used by the backward pass of previous layers, and ultimately to derive updates to trainable parameters.

The key issue in designing a backward pass for SMTLayer is the geometry of the functions computed by either forward pass. For any vector $v \in \{-1, 0, 1\}^p$ and $x, x'$ with $\mathrm{sign}(x) = \mathrm{sign}(x') = v$, then $\mathsf{B}_?^{\phi}(x) = \mathsf{B}_?^{\phi}(x')$, so these functions are piece-wise constant step functions ranging over the corners of the $\mathbb{R}^q$ unit hypercube. Thus, while they are differentiable almost everywhere, the gradient is not helpful for training because it is always zero. Prior work

---

**Algorithm 2:** $\mathsf{B}_{core}^{\phi}(z, y, \partial_y \ell(y, y^\star))$
unsat core-based backward pass

**Inputs:** $z \in \mathbb{R}^p$ input of forward pass
$\quad\quad y \in \mathbb{R}^q$ output of forward pass
$\quad\quad \partial_y \ell(y, y^\star)$ gradient with respect to output
$\quad\quad \phi(z_0, \ldots, z_{p-1}, y_0, \ldots, y_{q-1})$ a $T$-formula.
**Output:** $\partial_z \ell(y, y^\star) \in \mathbb{R}^p$ approximate gradient of $\ell$

1 **begin**
2 $\quad\quad G_z \leftarrow \partial_z \ell(z, \mathrm{sign}(z))$
3 $\quad\quad \hat{y} = \mathrm{sign}(y) - 2 \cdot \mathrm{sign}\left(\partial_y \ell(y, y^\star)\right)$
4 $\quad\quad$ **if** $\mathrm{sign}(y) \neq \mathrm{sign}(\hat{y})$ **then**
5 $\quad\quad\quad\quad z_b \leftarrow [z[i] > 0 \ : \ i = 0 \ldots p-1]$
$\quad\quad\quad\quad \hat{y}_b \leftarrow [\hat{y}[i] > 0 \ : \ i = 0 \ldots q-1]$
6 $\quad\quad\quad\quad \phi_z, \phi_y \leftarrow \bigwedge_{0 \leq i < p} z_i = z_b[i], \ \bigwedge_{0 \leq i < q} y_i = \hat{y}_b[i]$
7 $\quad\quad\quad\quad I \leftarrow \arg\min_{I \subseteq [0, p)} \mathbb{1}(\neg\phi \vee \neg\phi_y \vee \bigvee_{i \in I} z_i \neq z_b[i]) \cdot |I|$
8 $\quad\quad\quad\quad$ **foreach** $i \in I$ **do**
9 $\quad\quad\quad\quad\quad\quad G_z[i] \leftarrow \partial_{z[i]} \ell(z[i], 1 - \mathrm{sign}(z[i]))$
10 $\quad\quad\quad\quad$ **end**
11 $\quad\quad\quad\quad$ **foreach** $i \in \bar{I}$ **do**
12 $\quad\quad\quad\quad\quad\quad G_z[i] \leftarrow 0$
13 $\quad\quad\quad\quad$ **end**
14 $\quad\quad$ **end**
15 $\quad\quad$ **return** $G_z$
16 **end**

---

on integrating such functions into deep networks [35, 34] addresses this problem by relaxing the function computed by the forward pass, so that its gradients are hopefully more informative.

In contrast, $\mathsf{B}_{core}^{\phi}$ (Algorithm 2) and $\mathsf{B}_{max}^{\phi}$ (Algorithm 4 in Appendix A.2) do not attempt to provide gradients for a relaxation of the forward pass. Instead, they use information provided by the solver in its computation of the forward pass to identify which components of the input may have contributed to higher loss. The gradient is then computed by constructing a variant of the input provided to the forward pass, which differs on the identified components, and returning the gradient of the BCE loss of the original input on this variant. The two algorithms differ in the information that they extract from the solver, i.e., either solutions to a MaxSMT instance or an unsatisfiable core.

Both algorithms begin by initializing the gradient to be the loss between the logit inputs, and their hard labels (line 2). Recall that we assume the loss $\ell$ is binary cross-entropy, so the result will not

be zero. The purpose of this initialization is to emulate the dynamics of training with cross-entropy loss with a conventional layer; when the rounded output matches the target, the loss is not zero, and training will continue to move the parameters in a direction that makes them agree "more" with the hard target.

One line 3, they then use the provided gradient from the subsequent layer together with the memoized output from the forward pass to construct $\hat{y}$, a "corrected" output that satisfies $\ell(\hat{y}, y^\star) \leq \ell(y, y^\star)$. To understand why, observe that the sign of $\hat{y}$ computed on line 3 of both algorithms must be equal to that of $y^\star$. This follows from two facts: (1) at any coordinate $i$ where $y[i] \neq y^\star[i]$, $\mathsf{sign}(\partial_y \ell(y, y^\star))[i] = \mathsf{sign}(y)[i]$; (2) at any coordinate $i$ where $y[i] = y^\star[i]$, $\mathsf{sign}(\partial_y \ell(y, y^\star))[i] = -1 \cdot \mathsf{sign}(y)[i]$. If the sign of $\hat{y}$ is the same as that of $y$, then both algorithms return the initialized gradient. Otherwise, they extract information from the solver using $z$ and $\hat{y}$.

Algorithm 2 that runs $\mathsf{B}_{core}^\phi$ identifies a set of constraints $z_i = z_b[i]$ that are inconsistent with $\phi \wedge \phi_y$. Note that line 7 specifies a minimal unsatisfiable core, but this is not necessary. All that is needed is that none of the clauses in the core be superfluous, i.e., deleting any singleton clause from $I$ will cause it to be satisfiable. If a superfluous clause remains in the core, then the gradient returned for the corresponding input will have the incorrect sign, which may lead to issues with training. $\mathsf{B}_{core}^\phi$ then updates the gradient at each input identified in the core using the loss of $z[i]$ with respect to $1 - \mathsf{sign}(z[i])$, which will lead to updates in a direction that would have modified the input such that $i$ was not in the unsat core. The indices not in the unsat core have their gradients set to zero, as their absence in the core is not evidence that these inputs were correct or incorrect.

Algorithm 4 (in Appendix A.2) that runs $\mathsf{B}_{max}^\phi$ instead constructs a set of clauses $\phi_y$ that constrain the free $y_0, \ldots, y_{q-1}$ to take the values of $\hat{y}_b$, the Boolean conversion of $\hat{y}$. It then computes the softmax values of the absolute incoming logits $|z|$, and uses them to find the maximally-weighted set of clauses $(\mathsf{softmax}(|z|)[i], z_i = z_b[i])$ that are consistent with $\phi \wedge \phi_y$. Intuitively, these are the inputs that the previous layer is most confident in that can be made consistent with the corrected label $\hat{y}$ by changing some of the less confident inputs. $\mathsf{B}_{max}^\phi$ then updates the initialized gradient at each index for which the solution to this instance does not match the sign of the original input.

### 4.3 Analysis

To understand the settings where SMTLayer will provide optimal results, we introduce a class of *decomposable* learning problems (Definition 1).

**Definition 1** (Decomposable problem). Let $T$ be a first-order theory with constants in $\mathbf{Z}$. An ERM problem $\mathcal{D}, \mathcal{H}$ is *decomposable* by $T$ if there exists a function $f : \mathbf{X} \to \mathbf{Z}$, companion hypothesis class $\mathcal{H}_f \subseteq \mathbf{X} \to \mathbf{Z}$, and $T$-formula $\phi$ such that:

1. For any $h \in \mathcal{H}$, there exists $h_f \in \mathcal{H}_f$ and $h'$ such that $h = h' \circ h_f$.

2. There exists a random function $g : \wp(\mathbf{Y}) \to \mathbf{Y}$ such that for any $n > 0$ and $\forall S$ in the support of $\mathcal{D}^n$,
$$\Pr_{(x,y)\sim\mathcal{D}}[(x,y) \in S] = \Pr_{(x,\cdot)\sim\mathcal{D}}[(x, g(\langle x \rangle_{f,\phi})) \in S]$$
where $\langle x \rangle_{f,\phi} = \{y : \phi(f(x), y) \text{ is satisfied}\}$.

In (2), $f$ is called the *grounding function* and $\phi$ is called the *prediction logic*.

Intuitively, a learning problem defined in terms of a distribution $\mathcal{D}$ and hypothesis class $\mathcal{H}$ is decomposable if members of $\mathcal{H}$ can be decomposed into functions that are responsible for grounding and prediction, and $\mathcal{D}$ can be expressed in terms of a grounding function and a first-order formula $\phi$. There are a few important things to note. First, there is no requirement that the grounding function $f$ be a member of $\mathcal{H}_f$. While this may be realized at times, we should not assume that the data is actually generated by a function in the class that one learns over. In fact, we do not assume that $f$ is efficiently computable, as it may correspond to a natural process, or an aspect of data generation that is not understood well enough to make computational claims.

Second, for a given $x$, there may be more than one satisfying assignment for $y$ to $\phi(f(x), y)$. The function $g$ in (2) accounts for this, requiring only that when solutions to $\phi(f(x), y)$ are sampled by $g$, the result is distributed identically to $\mathcal{D}$. This paper will focus on the case where satisfying

assignments for $y$ are unique, as these are more in line with "classic" ERM classification problems. We leave exploration of the more general setting to future work.

We note that if the grounding function is known, can be computed efficiently, and $\phi$ is efficiently solvable, then the learning problem effectively has a closed-form solution. Rather, we assume that only $\phi$ and perhaps $g$ are known, and a sample of $\mathcal{D}$ is given. The remaining challenge is to identify a grounding hypothesis $h_f \in \mathcal{H}_f$ for which the construction in (2) is an effective solution to the end-to-end learning problem posed by $\mathcal{D}, \mathcal{H}$. This stands in contrast to traditional ERM, in which a good solution $h \in \mathcal{H}$ must either solve both grounding and prediction, or find a "shortcut" that manages to predict $\mathcal{D}$ as well as the decomposition.

**Convergence.** Regarding the backward passes, Theorem 2 below demonstrates that when $\phi$ satisfies certain conditions, and the companion hypothesis class $\mathcal{H}_f$ satisfies conditions that are sufficient to guarantee convergence with SGD, then training with $\mathsf{F}^{\phi}_{smt}$ and $\mathsf{B}^{\phi}_{max}$ will converge to the optimal solution in the number of iterations. The proof of this theorem is based on the observation that when the conditions on $\phi$ are met, then training with $\mathsf{B}^{\phi}_{max}$ obtains the same solution that would be obtained if the labels of $\phi$ were available for supervised learning, so it is possible to use the solver's solutions interchangeably with the correct supervised labels. Thus, the conditions on $\mathcal{H}_f$ are sufficient to ensure the stated convergence, as stated in a well-known result outlined in Chapter 14 of [26].

It is also worth noting that Theorem 2 does not necessarily hold if $\mathsf{B}^{\phi}_{core}$ is used instead of $\mathsf{B}^{\phi}_{max}$. The reason is that there may be many unsatisfiable cores that are locally minimal in cardinality, and gradients are set only for inputs that appear in the computed core. These gradients will not match those of the loss on a grounding sample, so the training dynamics are likely to be different. We believe that training with $\mathsf{B}^{\phi}_{core}$ may have more in common with block coordinate descent than gradient descent, and save a more detailed exploration of the topic for future work.

**Theorem 2.** *Let $\mathcal{D}, \mathcal{H}$ be a $T$-decomposable problem with grounding function $f$ and prediction logic $\phi$ where:*

1. *$\mathbf{Z}$ and $\mathbf{Y}$ are Cartesian products of Booleans.*

2. *For any $(x, y) \sim \mathcal{D}$ and $y' \neq y$, $\phi(f(x), y')$ is $T$-equivalent to false and there is exactly one $z$ such that $\phi(z, y)$ is $T$-equivalent to true.*

3. *$\mathcal{H}_f$ is a convex set and for all $h_f \in \mathcal{H}_f$, $\|h_f\| \leq B$, and the loss $\ell(h_f(\cdot), z)$ is $M$-Lipschitz and convex in $x$ for any fixed $z$.*

*Then for any $\epsilon > 0$, selecting $h_f$ by minimizing either $L_S(\mathsf{F}^{\phi}_{smt}(h_f(\cdot)))$ with $\tau \geq M^2 B^2 / \epsilon^2$ iterations of stochastic gradient descent, with gradients provided by $\mathsf{B}^{\phi}_{max}$, and learning rate $\eta = \sqrt{B^2 / M^2 \tau}$ yields a grounding hypothesis $\hat{h}_f \in \mathcal{H}_f$ that satisfies: $\mathbb{E}[L_{\mathcal{D}}(\hat{h}_f)] \leq \min_{h_f \in \mathcal{H}_f} L_{\mathcal{D}}(h_f) + \epsilon$. The randomness in this expectation is taken over the choices of the SGD algorithm.*

## 5   Experimental Evaluation

In this section we present an empirical evaluation of SMTLayer on four learning problems that can be decomposed into perceptual and symbolic subtasks. Our results demonstrate the following primary findings. *1)* SMTLayer is effective: on every benchmark, it provides superior results over "conventional" learning that takes place without encoded symbolic knowledge. *2)* SMTLayer has distinct advantages over prior approaches. Compared with SATNet [35], it requires *significantly* less training data to converge, and in all cases yields a more accurate model; compared with Scallop [13], it is less computationally expensive, requires less training data, and it is more expressive in terms of the knowledge that it can encode; compared to approaches that incorporate symbolic knowledge into training, but do not use it during inference [1], SMTLayer gives superior results on test data. *3)* Models trained with SMTLayer may be more robust to certain types of covariate shift that occur relative to the symbolic component of the problem; when SMTLayer succeeds at learning a representation, it will continue to produce correct inferences provided the perceptual component remains stationary.

## 5.1 Datasets

Additional details on the datasets and corresponding architectures used in our evaluation can be found in Appendix A.4. Hyper-parameters used for training are in Appendix A.5.

**MNIST Addition.**    The MNIST addition problem is illustrated in Figure 1, and is similar to the benchmark described by [13]. For training, we use "MNIST +$p$%" to denote a training set of size 60,000 that contains $p$% of the possible pairs of digits. So $p = 100$ indicates all possible pairs of digits are used, and for $p = 10$, we only use pairs of the same digit. We use $p = 10, 25, 50, 75$ and 100 in our experiments. In all cases, we use the same test set of all possible pairs of digits.

**Visual Algebra.**    The task is to solve for the variable $x$ in a graphical depiction of the equation $ax + b = c$, where $a, b$ and $c$ are randomly-chosen numbers, and each symbol is depicted visually using EMNIST [5] and HASY graphics [29]. Similar to MNIST addition, the training sample selects $a$ and $b$ uniformly from pairs of the same digit, and $x$ uniformly from the odd numbers between 0 and 9. The test sample was generated by sampling $a, b$, and $x$ uniformly.

**Liar's Puzzle.**    The liar's puzzle is comprised of three sentences spoken by three distinct agents: Alice, Bob, and Charlie. One of the agents is "guilty" of an unspecified offense, and in each sentence, the corresponding agent either states that one of the other parties is either guilty or innocent. It is assumed that two of the agents are honest, and the guilty party is not. The objective is to identify the guilty party. A formal characterization of the underlying logic is given in Appendix A.4. We note that the logic has non-stratified occurrences of negation, so it cannot be encoded with Scallop.

**Visual Sudoku.**    This task is to complete a $9 \times 9$ Sudoku board where each entry is an MNIST digit. We use the dataset from the SATNet evaluation [35], and examine three configurations obtained by sampling 10%, 50%, and 100% of the original training set. Although there are examples of Sudoku solvers implemented as logic programs, we were not able to implement one in Scallop without violating stratified negation. When calculating accuracy, we check if the *entire* board is correct.

## 5.2 Setup

We implemented a prototype of our approach using Pytorch [22] and Z3 [7], which will be made available in open-source when this paper is published. When training models with SMTLayer, we use SGD with Nesterov momentum at rate 0.9 and gradient clipping rate 0.1. Before training a model with SMTLayer (or a comparison technique, unless stated otherwise), we first pre-train the neural network by replacing SMTLayer with a dense network containing one hidden layer of 512 neurons. This can potentially limit the number of training updates needed at lower layers, but will not result in a model with a representation that is compatible with symbolic knowledge, so further training is needed. Results in Table 1 (left) were averaged over five runs of training. Results in Table 1 (right) for SATNet [35] and semantic strengthening (Ahmed et al. [1]) were reported in the original papers; details on our fine-tuning of GPT2-XL are given in Appendix A.6.

## 5.3 Results

**Overall performance.**    In terms of accuracy, Table 1 (left) shows that SMTLayer outperforms prior work in terms of accuracy, training time, or both, on all configurations. SMTLayer is consistently faster than Scallop, nearly $4\times$ in the case of visual algebra. The per-epoch time to train the SATNet models is less expensive than SMTLayer, but this is not always conclusive. In the case of visual sudoku, the 10% SMTLayer model achieved superior error rates in 20 epochs, compared with 100 epochs for the 100% SATNet model. We also point out that although Theorem 2 suggests that Algorithm 4 might have learning advantages over Algorithm 2 found this not to be the case on these datasets. All of the results in Table 1 (left) were trained with Algorithm 2, and test inference was done using Algorithm 1.

**Solvers during inference.**    Table 1 (right) shows the performance of several approaches for solving (non-visual) Sudoku puzzles presented as text. The purpose of this comparison is to highlight the advantages of using symbolic information during inference, as done by SMTLayer, SATNet, and Scallop, versus only during training, as in the case of Semantic Strengthening [1]. We include several

| configuration | SMTLayer | | SATNet [35] | | Scallop [13] | |
|---|---|---|---|---|---|---|
| | test acc.(%) | epoch time(sec.) | test acc.(%) | epoch time(sec.) | test acc.(%) | epoch time(sec.) |
| MNIST+ 10% | **98.1** | 75.4 | 10.0 | 31.0 | 33.7 | 96.3 |
| MNIST+ 25% | **98.3** | 74.8 | 34.2 | 30.9 | 65.8 | 96.4 |
| MNIST+ 50% | **98.6** | 75.8 | 54.8 | 32.8 | 98.4 | 96.5 |
| MNIST+ 75% | **98.5** | 75.0 | 78.4 | 31.9 | 93.5 | 96.4 |
| MNIST+ 100% | **98.5** | 75.8 | 96.7 | 33.5 | 98.6 | 96.6 |
| Vis. Alg. #1 | **98.2** | 168.2 | 19.6 | 80.1 | 18.7 | 602.8 |
| Vis. Alg. #2 | **25.4** | 127.2 | 18.6 | 52.5 | 21.3 | 636.1 |
| Liar's Puzzle | **86.1** | 28.7 | 84.6 | 3.0 | — | — |
| Vis. Sudoku 10% | **66.0** | 135.7 | 0.0 | 9.9 | — | — |
| Vis. Sudoku 50% | **73.1** | 608.1 | 0.0 | 45.4 | — | — |
| Vis. Sudoku 100% | **79.1** | 1199.0 | 63.2 | 86.5 | — | — |

| approach | correct (%) | time (sec) |
|---|---|---|
| SMT | 100.0 | 0.05 |
| SATNet [35] | 98.3 | 0.01 |
| Ahmed et al. [1] | 28.0 | 0.01 |
| gpt2-xl-9K @ 1 | 2.3 | 2.22 |
| gpt2-xl-9K @ 10 | 6.6 | 2.36 |
| gpt2-xl-9K @ 100 | 11.1 | 4.99 |
| gpt2-xl-1M @ 1 | 14.3 | 2.16 |
| gpt2-xl-1M @ 10 | 39.8 | 2.29 |
| gpt2-xl-1M @ 100 | 66.1 | 4.52 |
| text-davinci-003 @ 1 | 0.0 | 12.56 |
| gpt-3.5-turbo-0301 @ 1 | 0.3 | 19.28 |

Table 1: **(Left)** Results after training and inference with SMTLayer versus prior work. All SMTLayer tests were measured with the MaxSMT forward pass. Epoch times are averaged over all epochs on which the model was trained. — cells denote incompatibility with the approach. **(Right)** Comparison with other neuro-symbolic and transformer-based approaches on the plain (non-visual) Sudoku benchmark from Wang et al [35]. GPT2-XL was fine-tuned on either the training portion of the benchmark (*9K*), or 1 million random instances (*1M*). Times reflect completion of one puzzle (using OpenAI's API for davinci and turbo-3.5), and @ is the number of attempts generated.

transformer-based approaches in this comparison, as there is widespread interest in their ability to perform this type of reasoning, despite not making explicit use of symbolic information. Although Semantic Strengthening and transformer-based approaches could be used to solve Visual Sudoku (in the latter case, via a vision encoder-decoder), we are not aware of a public implementation of either, so these numbers represent an upper-bound on what their performance on that benchmark would be.

The top row shows the performance of Z3 on the benchmark test set, and is included as a baseline for the time needed to solve a puzzle. SATNet, which includes a differentiable relaxation of a SAT solver, clearly performs the best of the learning-based approaches, and beats SMTLayer's performance of Vis. Sudoku because there is no possibility of grounding error. While it still must learn the rules of Sudoku, it has a strong prior that enables it to do so very well; on the other hand, Semantic Strengthening leverages Sudoku constraints during training, but does not learn a complete neural representation of them to use during inference. The OpenAI models can recite the rules of Sudoku when prompted, but are rarely successful in applying them to a specific puzzle (see Appendix A.6 for details). The GPT2-XL models tuned on the 9K benchmark instances do better, but it is only when they are tuned on *significantly* more data (1M) that they begin to approach acceptable levels of performance. It is important to note that this is only true when given 100 attempts, making them more costly than approaches that leverage solvers.

**Training sample size.** Because SMTLayer encodes explicit knowledge that is essential to correct inference on these datasets, our approach is able to perform well in data-impoverished settings where the training sample is insufficient to fully specify the symbolic component of the learning task. This is readily apparent across the results in Table 1: in the MNIST addition and first visual algebra configuration, SMTLayer yields a model that performs nearly perfectly despite not being given a sufficient sample in most cases. Because SATNet must learn the symbolic component, it is at a disadvantage, and in these settings performs similarly to a conventional model. In theory, Scallop should be able to perform as well as SMTLayer, as it also encodes explicit knowledge. However, it is unable to learn a useful model for either visual algebra configuration, and does not learn the correct representation for MNIST addition until it sees half of the possible digit pairs during training.

SMTLayer does particularly well on the visual Sudoku dataset introduced by [35]. When trained on just 10% of the original sample, it learns a function that *exceeds* the performance of the SATNet model by a healthy margin, which continues to grow as it is exposed to more training data. On the other hand, we found that SATNet failed to converge with less than the full original training sample.

**Robustness & interpretability.** The reason that SMTLayer is able to perform well, and often near the optimum, in configurations that other approaches perform poorly on, is that it learns a representation that is consistent with the symbolic knowledge encoded in the SMTLayer. For example, the constraints that we use for MNIST addition, visual algebra, and visual sudoku all encode

digits as bitvectors. In order to make a correct inference, the neural network must learn to encode MNIST digits in their correct bitvector representation. If learning succeeds at this, then there are two positive outcomes that follow. First, the model's representation will be inherently *interpretable*, because it will coincide with the provided symbolic domain knowledge. Second, the resulting model is naturally robust to covariate shift that does not affect the distribution of perceptual data that the network grounds, but that does affect the statistics of their composition.

This type of shift is on display in the MNIST 10% and visual algebra experiments, where at training time, the model only sees pairs of same-numbered digits, and at test time it is exposed to a substantially different distribution of digit pairs or formulas. We verified this by examining the representations learned by SMTLayer and Scallop on MNIST Addition 10%; it is unreasonable to expect SATNet to learn an interpretable representation. As expected, SMTLayer produces the correct representation in proportion to the accuracy of a typical MNIST model ($\approx 99\%$), whereas Scallop's representation was correct roughly 50% of the time. However, architecture plays a role in this robustness, as shown in the SMTLayer results for the second visual algebra configuration. Because the network is shown the full instance, and not the individual digits, it learns the training bias. Despite having access to the symbolic formulas in SMTLayer, it cannot disentangle the perceptual symbols from their covariance.

## 6   Limitations

A major constraint imposed by our approach arises from the SMTLayer's dependence on Boolean vectors as a means to interact with the theoretical framework. This compels the conversion of continuous values into discrete forms, a process that risks compromising inference accuracy and adding unnecessary complexity to the theoretical underpinnings. The integration of SMTLayer also limits the amount of parallelism that can be used during training and inference, as the SMT solver must be run on CPU cores. While it may be possible with significant engineering effort to move some of this functionality to GPU cores, there will likely always be a need to run portions of SMTLayer on general-purpose CPU cores, raising the cost and decreasing efficiency in batched settings.

## 7   Conclusion

Our approach for integrating logical theories into deep learning, SMTLayer, provides a pragmatic solution to the problem of incorporating symbolic knowledge into learning for training and inference, which we demonstrate on several problems involving both perceptual tasks—vision and natural language—and logical reasoning. Notably, we show that models which incorporate symbolic knowledge during training and inference can outperform conventional models as well as prior work in this area, especially in settings where training data is limited. Continued progress on automated reasoning techniques has played a pivotal role in the development of several fields over the past decades, and our hope is that the contributions in this paper will aid in progress towards realizing their potential in challenges that surpass the capabilities of existing learning techniques.

**Acknowledgements.**   This work was supported by the U.S. Army Research Office under MURI Grant W911NF-21-1-0317, and the National Science Foundation under Grant No. CNS-1943016.

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
