# A Appendix

## A.1 Broader Impacts

This research integrates SMT solvers into Deep Neural Networks, resulting in models that potentially require fewer training samples, are more robust, and produce interpretable outputs. While these advancements are promising, there are also potential misuses, ethical implications, and long-term effects to consider. As with any AI technology, there is a potential for misuse. The models resulting from this work could be used in systems where decisions have significant impacts on individuals' lives. If not properly regulated or used responsibly, they could potentially perpetuate or even exacerbate existing biases or inequalities. Additionally, the increased interpretability of these models could be misused to more effectively manipulate or persuade individuals, contributing to unwanted targeting or misinformation. The ethical implications of this work largely stem from its potential misuses. Ensuring fair, accountable, and transparent use of these models is crucial. It's also necessary to ensure that the symbolic knowledge integrated into these models accurately represents the problem space and doesn't incorporate biases or harmful stereotypes. In the long term, by bridging the gap between symbolic reasoning and inductive learning, we could see the development of more capable systems. However, the potential for misuse and ethical implications could also have long-term societal impacts if not properly addressed. As AI becomes more prevalent, the potential for harm due to bias, lack of transparency, or misuse increases. It's essential to have ongoing discussions about the regulation and ethical use of AI to mitigate these potential long-term effects.

## A.2 Algorithms

In this section we present the MaxSMT-based variants of our forward and backward passes (Algorithms 3 and 4). Both of the algorithms are parameterized by a user-provided first-order formula $\phi$, and take a single vector-valued input consisting of unscaled floating-point values (*logits*). These values are cast to Boolean constants by taking their sign on line 2 of both algorithms, so that they can be equated with the corresponding free variables $z_0, \ldots, z_{p-1}$.

The key difference between $\mathsf{F}_{max}^{\phi}$ and $\mathsf{F}_{smt}^{\phi}$ is the way in which they handle inputs that are inconsistent with $\phi$ when interpreted as Booleans. $\mathsf{F}_{smt}^{\phi}$ addresses this by providing an output that is also inconsistent with $\phi$, i.e. a vector of zeroes, effectively signaling that the network below it did not provide consistent inputs. Alternatively, we can interpret the values provided by the network as Booleans enriched with "confidence" values. Although we expect inputs to $\mathsf{SMTLayer}$ to be unscaled floating-point values, Algorithm 3 scales them to a formal probability distribution via the softmax function for use as weights to find the weighted MaxSMT solution of $\phi$. With this approach, $\mathsf{SMTLayer}$ will always provide a valid, although not necessarily correct, output that is consistent with the inputs of which the network below is most "confident".

Regarding the backward pass, neither $\mathsf{B}_{core}^{\phi}$ nor $\mathsf{B}_{max}^{\phi}$ attempt to provide gradients for a relaxation of the forward pass. Instead, they use information provided by the solver in its computation of the forward pass to identify which components of the input may have contributed to higher loss. The gradient is then computed by constructing a counterfactual variant of the input provided to the forward pass, which differs on the identified components, and returning the gradient of the binary cross-entropy loss of the original input on this counterfactual variant. The two algorithms differ in the information that they extract from the solver, i.e., either solutions to a MaxSMT instance or an unsatisfiable core.

$\mathsf{B}_{max}^{\phi}$ constructs a set of clauses $\phi_y$ that constrain the free $y_0, \ldots, y_{q-1}$ to take the values of $\hat{y}_b$, the Boolean conversion of $\hat{y}$. It then computes the softmax values of the absolute incoming logits $|z|$, and uses them to find the maximally-weighted set of clauses $(\mathsf{softmax}(|z|)[i], z_i = z_b[i])$ that are consistent with $\phi \wedge \phi_y$. Intuitively, these are the inputs that the previous layer is most confident in that can be made consistent with the corrected label $\hat{y}$ by changing some of the less confident inputs. $\mathsf{B}_{max}^{\phi}$ then updates the initialized gradient at each index for which the solution to this instance does not match the sign of the original input.

**Algorithm 3:** $\mathsf{F}^\phi_{max}(z)$

MaxSMT-based forward pass of SMTLayer

---

**Inputs:** $z \in \mathbb{R}^p$ layer input
$\qquad\quad \phi(z_0, \ldots, z_{p-1}, y_0, \ldots, y_{q-1})$ $T$-formula
**Output:** $y \in \mathbb{R}^q$

1 **begin**
2 $\quad z_b \leftarrow [z[i] > 0 \; : \; i = 0 \ldots p-1]$
3 $\quad C \leftarrow \sum_{i \in I} \mathsf{softmax}(|z|)[i]$
4 $\quad y \leftarrow \arg\max_{y_b} \max_I C \cdot \mathbb{1}(\phi \wedge \bigwedge_{i \in I} y_i = y_b[i] \wedge z_i = z_b[i])$
5 $\quad$ **return** $y$
6 **end**

---

**Algorithm 4:** $\mathsf{B}^\phi_{max}(z, y, \partial_y \ell(y, y^\star))$

MaxSMT-based backward pass of SMTLayer

---

**Inputs:** $z \in \mathbb{R}^p$ input of forward pass
$\qquad\quad y \in \mathbb{R}^q$ output of forward pass
$\qquad\quad \partial_y \ell(y, y^\star)$ gradient with respect to output
$\qquad\quad \phi(z_0, \ldots, z_{p-1}, y_0, \ldots, y_{q-1})$ a $T$-formula
**Output:** $\partial_z \ell(y, y^\star) \in \mathbb{R}^p$ approximate gradient of $\ell$

1 **begin**
2 $\quad G_z \leftarrow \partial_z \ell(z, \mathsf{sign}(z))$
3 $\quad \hat{y} = \mathsf{sign}(y) - 2 \cdot \mathsf{sign}\left(\partial_y \ell(y, y^\star)\right)$
4 $\quad$ **if** $\mathsf{sign}(y) \neq \mathsf{sign}(\hat{y})$ **then**
5 $\quad\quad z_b \leftarrow [z[i] > 0 \; : \; i = 0 \ldots p-1]$
6 $\quad\quad \hat{y}_b \leftarrow [\hat{y}[i] > 0 \; : \; i = 0 \ldots q-1]$
7 $\quad\quad \phi_y \leftarrow \bigwedge_{0 \leq i < q} y_i = \hat{y}_b[i]$
8 $\quad\quad C \leftarrow \sum_{i \in I} \mathsf{softmax}(|z|)[i]$
9 $\quad\quad I \leftarrow \arg\max_{I \subseteq [0,p)} \mathbb{1}(\phi \wedge \phi_y \wedge \bigwedge_{i \in I} z_i = z_b[i]) \cdot C$
10 $\quad\quad$ **foreach** $i \in \bar{I}$ **do**
11 $\quad\quad\quad G_z[i] \leftarrow \partial_{z[i]} \ell(z[i], 1 - \mathsf{sign}(z[i]))$
12 $\quad\quad$ **end**
13 $\quad$ **end**
14 $\quad$ **return** $G_z$
15 **end**

## A.3 Proofs

**Theorem 2.** *Let $\mathcal{D}, \mathcal{H}$ be a $T$-decomposable problem with grounding function $f$ and prediction logic $\phi$ where:*

    1. $\mathbf{Z}$ *and* $\mathbf{Y}$ *are Cartesian products of Booleans.*

    2. *For any $(x, y) \sim \mathcal{D}$ and $y' \neq y$, $\phi(f(x), y')$ is $T$-equivalent to false and there is exactly one $z$ such that $\phi(z, y)$ is $T$-equivalent to true.*

    3. $\mathcal{H}_f$ *is a convex set and for all $h_f \in \mathcal{H}_f$, $\|h_f\| \leq B$, and the loss $\ell(h_f(\cdot), z)$ is $M$-Lipschitz and convex in $x$ for any fixed $z$.*

*Then for any $\epsilon > 0$, selecting $h_f$ by minimizing either $L_S(\mathsf{F}^\phi_{smt}(h_f(\cdot)))$ with $\tau \geq {}^{M^2 B^2}/_{\epsilon^2}$ iterations of stochastic gradient descent, with gradients provided by $\mathsf{B}^\phi_{max}$, and learning rate $\eta = \sqrt{B^2/M^2\tau}$ yields a grounding hypothesis $\hat{h}_f \in \mathcal{H}_f$ that satisfies: $\mathbb{E}[L_\mathcal{D}(\hat{h}_f)] \leq \min_{h_f \in \mathcal{H}_f} L_\mathcal{D}(h_f) + \epsilon$. The randomness in this expectation is taken over the choices of the SGD algorithm.*

*Proof.* To prove this result, we introduce the notion of a *grounding sample*.

**Definition 3** (Grounding sample)**.** Let $\mathcal{D}, \mathcal{H}$ be a $T$-decomposable problem with grounding function $f$. The grounding sample $S_f$ for $S \sim \mathcal{D}$ is given by $[(x_i, f(x_i)) : (x_i, y_i) \in S]$, i.e., tuples that consist of the first element of each instance in $S$ and its image under $f$.

Now observe that the conditions stated in assumptions (1) and (3) are sufficient to yield the result if instead of optimizing $L_S(\mathsf{F}^\phi_{smt}(h_f(\cdot)))$, we were given the grounding sample $S_f$ and minimized $L_{S_f}(h_f)$ (see [26], Theorem 14.8). The result follows as stated then because of assumption (2), which implies that the update vectors provided by $\mathsf{B}^\phi_{max}$ are the gradients of $L_{S_f}(h_f)$.

To understand why, observe that the sign of $\hat{y}$ computed on line 3 of both algorithms must be equal to that of $y^\star$. This follows from two facts:

1. At any coordinate $i$ where $y[i] \neq \mathsf{y}^\star[i]$, $\mathsf{sign}(\partial_y \ell(y, y^\star))[i] = \mathsf{sign}(y)[i]$.

2. At any coordinate $i$ where $y[i] = y^\star[i]$, $\mathsf{sign}(\partial_y \ell(y, y^\star))[i] = -1 \cdot \mathsf{sign}(y)[i]$.

Now there are two cases to consider.

**Case 1:** $\mathsf{sign}(y) = \mathsf{sign}(\hat{y})$. In this case, the algorithm returns $\partial_z \ell(z, \mathsf{sign}(z))$. Because solutions to $\phi(\cdot, y)$ are unique, $\mathsf{sign}(z)$ is the correct grounding, i.e., $z = f(x)$ for the original features $x$.

**Case 2:** $\mathsf{sign}(y) \neq \mathsf{sign}(\hat{y})$. Because of assumption (2), the set of indices computed on line 8 will contain all of the coordinates at which $z$ matches the correct value $z^\star = f(x)$. Note that at these coordinates, the vector returned by the algorithm matches the gradient of $\ell(z, z^\star)$, which is $\partial_{z[i]} \ell(z[i], \mathsf{sign}(z[i]))$. In the remaining coordinates, the vector will contain $\partial_{z[i]} \ell(z[i], 1 - \mathsf{sign}(z[i]))$, which also matches the gradient of $\ell(z, z^\star)$.

The result follows. $\qquad\square$

### A.4 Dataset details

Two of the three problems that we examine are based on the MNIST handwritten digit dataset [8], which consists of 60,000 28x28 gray-scale images of handwritten numerals for training and 10,000 instances for testing. The digits on the left of Figure 1 are examples of instances from this data. To generate data for the visual algebra problem, we additionally drew from EMNIST [5], which extends MNIST with handwritten letters, and HASY [29], which contains handwritten symbols with similar characteristics to MNIST. For the liar's puzzle, inspired by examples [38] which formulate similar examples as SMT constraints, we constructed examples using a set of common phrases that we devised ourselves, and did not otherwise draw from publicly-available data.

Below, we describe the way in which we used these data sources to construct training and test samples, and the neural network architectures that we used with SMTLayer to solve them.

**MNIST Addition.** The MNIST addition problem is described in Example 1. In each instance, two MNIST digits are presented as features, and the task of the model is to provide their sum represented as a bitvector. The architecture that we use consists of four convolutional layers with kernels of size 3, depths in the order 64, 64, 128, 128, and a stride of width 2 on the first layer, and two dense layers of width 256 and 4. This network is applied to each digit, and the results are concatenated to obtain a vector of size 8 that is passed to an instance of SMTLayer with SMT constraints from Example 1, which ultimately produces a vector of width 5 that represents the bitvector sum of the digits.

We generated five training samples starting with one containing only pairs of the same digit, i.e. $(0,0), (1,1), \ldots$. We then added progressively more from the full set of possible pairs, using 25%, 50%, and 100%, and trained on batches of 128 across all datasets. Note that although we change the number of digit pairs that appear between samples, we always map these pairs to random MNIST images to obtain 60,000 training instances. This is to ensure that the training sample contains a sufficient sample of MNIST images to be able to perform well on the test data. In all cases, we use the same test set consisting of instances from all possible pairs of digits. The purpose of this is to demonstrate that the conventional network will not generalize until it has seen the full distribution, whereas the model with SMTLayer should be able to generalize after seeing many fewer examples.

**Visual Algebra.** The visual algebra problem is described in Example 2 and Example 3. Recall that features depict handwritten depictions of linear equations of the form $ax + b = c$. Values for $a, x$ and $b$ are randomly drawn from the range 1-9 to ensure that solutions are unique. Then the

corresponding value of $c$ is decomposed into $c = 10 \cdot c_1 + c_2$, and MNIST digits are selected at random to represent $a, b, c_1, c_2$. A random EMNIST alphabet character is drawn for the variable, and random multiplication, addition, and equality symbols are drawn from HASY. A minor note is that HASY does not contain the standard equality symbol "=", so we instead use "≐". These images are then concatenated horizontally in the appropriate order.

We evaluate two architectures for this problem. The first uses the same neural network that was used for MNIST addition, and an instance of SMTLayer with the constraints given in Example 2. It assumes that the four numeric digits in the problem have already been extracted, e.g. by a separate vision routine that can recognize digits from letters and arithmetic symbols, and are provided directly to the model. The four inputs are given separately to the neural network, which produces four 4-bit bitvectors that are concatenated and passed to SMTLayer, which produces a 4-bit bitvector result. We refer to this as configuration #1 in our results.

The second uses an architecture which takes the entire image containing the problem as a whole, and produces a 16-bit bitvector that is passed directly to SMTLayer. This architecture uses a similar stack of convolutional layers, but has a larger initial dense layer containing 26,112 neurons, as it is given a larger image. The difference in the convolutional stack is at the second layer, which also has a stride of width 2, to reduce the size of the feature map and mitigate the need for an even larger dense connection. The difference between these architectures relates to one of the challenges of this problem. Much of the information contained in the features is irrelevant to the solution, e.g., it is irrelevant which letter is chosen for the variable, or what the arithmetic operators look like, so this architecture must also learn to disregard these parts of the instance. We refer to this as configuration #2 in our results and in Example 3.

We generated a training sample by selecting $a$ and $b$ uniformly from pairs of the same digit, i.e. $(0, 0), (1, 1), \ldots$, and sampling $x$ uniformly from the odd numbers between 0 and 9. The test sample was generated by sampling $a, b$ uniformly from all pairs of digits, and $x$ from all numbers 0 to 9. We then map these values to random MNIST, EMNIST, and HASY images to obtain 60,000 samples. The intention is to study a problem wherein the model is not shown all possible problems (modulo representation as digits), or all of the solutions. This is more challenging than MNIST addition for two reasons: for a given solution, there are many more compatible ground terms, and the model does not see examples of some of the solutions it must provide for the test set. Thus, in order for SMTLayer to succeed, it must use the provided symbolic knowledge to approximate the correct grounding function, despite these deficiencies in the data.

**Liar's Puzzle.** The liar's puzzle is comprised of three sentences spoken by three distinct agents: Alice, Bob, and Charlie. One of the agents is "guilty" of an unspecified offense, and in each sentence, the corresponding agent either states that one of the other parties is either guilty or innocent. It is assumed that two of the agents are honest, and the guilty party is not. The solution to the problem is an identification of the guilty party. An example is described in Example 4

We synthesized a dataset for the liar's puzzle based on a limited set of utterances about who speaks in each sentence, which agent is the subject, and whether the subject is guilty or innocent. There are five ways of denoting the speaker: *"\* says"*, *"\* says that"*, *"\* said"*, *"\* said that"*, and a colon *"\* :"* separating the speaker's name from the rest of the sentence. There are five ways of uttering either innocence or guilt: *"\* did it/did not do it"*, *"\* is guilty/innocent"*, *"\* is definitely guilty/innocent"*, and *"\* definitely did it/did not do it"*, *"\* is the criminal/is a good person"*. We generated all of the combinations of subject, speaker, and proclaimed innocence or guilt, and took the product with all possible combinations of these utterances. The result is a dataset of 375,000 instances, each containing three natural language sentences.

The prediction logic for this problem assumes a set of ground predicates speaker($agent, sentence$), subject($agent, sentence$), accuses($sentence$), and guilty($agent$). For example, if the first sentence was *"Alice says that Bob is innocent"*, the ground predicates would be speaker(alice, 1), subject(bob, 1), and ¬guilty(bob). Then the prediction logic is shown in Equation 1.

$$
\begin{aligned}
& |\{a : \mathsf{guilty}(a)\}| = 1 \\
\wedge \quad & \forall a.|\{s : \mathsf{speaker}(a, s)\}| = 1 \\
\wedge \quad & \forall s.|\{a : \mathsf{subject}(a, s)\}| = 1 \\
\wedge \quad & \forall s, a_1, a_2.\mathsf{speaker}(a_1, s) \wedge \mathsf{subject}(a_2, s) \rightarrow \\
& \quad (\neg\mathsf{guilty}(a_1) \leftrightarrow \mathsf{accuses}(s) \leftrightarrow \mathsf{guilty}(a_2))
\end{aligned}
\tag{1}
$$

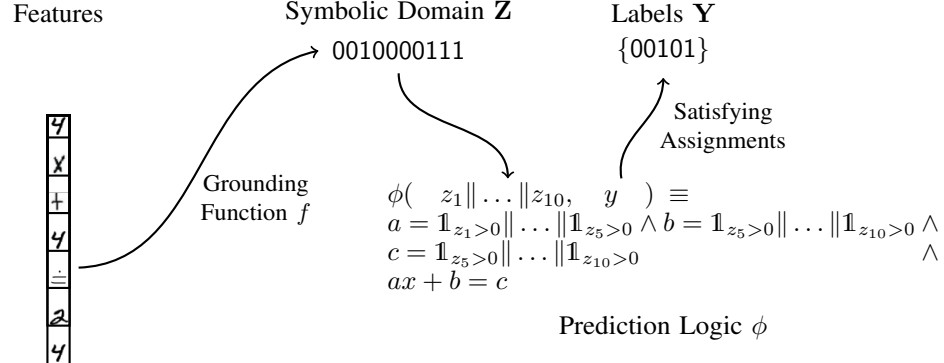

Figure 2: Visual Algebra configuration 1 example.

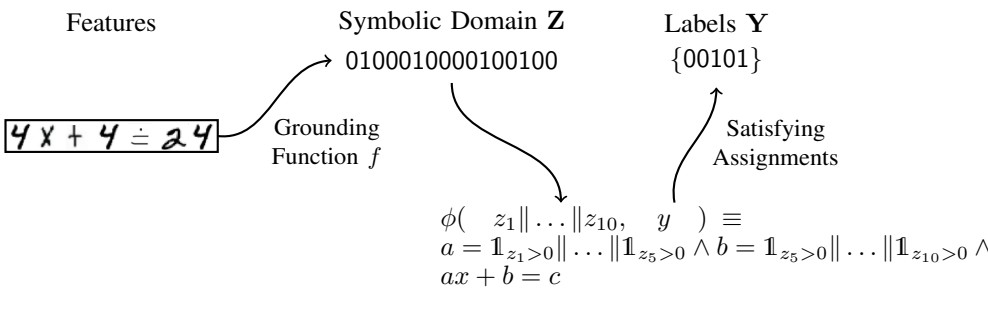

Figure 3: Visual Algebra configuration 2 example.

In our implementation, agents and sentence identifiers are encoded unary as 3-bit bitvectors. The quantifiers are removed by substituting for each ground term or sentence identifier, and the cardinality constraints are expanded into propositional formulas. The architecture that we adopt is a two-layer bidirectional long short-term model (LSTM) with 512 dimensions at each layer, and a 300-dimension trainable embedding layer initialized from GloVe-6B [23]. The hidden units of the last LSTM layer were connected to 2-layer dense network containing 128 followed by 7 neurons. This network is applied to each sentence in the input, and the concatenated results are passed to the SMTLayer, which solves the formula in Equation 1 to produce a unary encoding of the guilty party.

To evaluate solver layers on this problem, we selected training and test samples by first subsampling half of the full 375,000 available instances. We then selected half of the speaker, subject, accuses predicate configurations for all three sentences appearing in this subsample to appear in the training sample, and the other half to appear in the test sample. To further limit the amount of information in the training sample, we randomly selected one ordering for each predicate configuration to remain for training. There were 9,400 resulting training instances, and 28,062 test instances. Restricting the training set as described ensures that the model is trained on a limited subset of possible sentence configurations, and one that is logically disjoint from those that appear in the test sample. Because there is not enough information in the training sample to learn Equation 1, we expect only the model with SMTLayer to succeed, but to do so it must approximate the grounding function well from a limited sample.

### A.5 Hyperparameters

In our study we evaluate four problems: MNIST Addition, Visual Algebra, Liar's Puzzle and Visual Sudoku. They are diverse as covering both vision and language tasks, varying from a simple logical problem with fewer variables (e.g. addition) to the more complex one (Visual Sudoku). As the four problems that our evaluation studies vary considerably in size and complexity, the models used to train them require different considerations. This section details these differences. Table 2

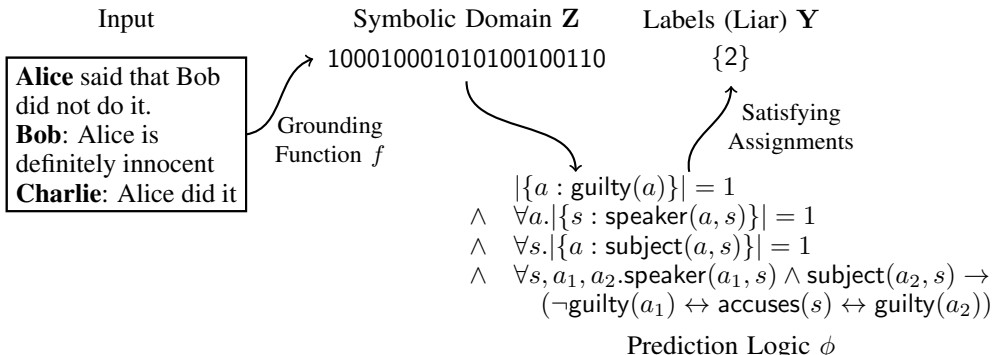

Figure 4: Liar's Puzzle example.

| | *Conventional* | | *w/* SMTLayer | | *w/* SATNet | | *w/* Scallop | |
|---|---|---|---|---|---|---|---|---|
| | *optimizer* | *epochs* | *optimizer* | *epochs* | *optimizer* | *epochs* | *optimizer* | *epochs* |
| MNIST+ 10% | SGD(1.0) | 0/5 | SGD(1.0) | 3/5 | Adam(2e-3, 1e-5) | 3/5 | Adam(1e-3) | 3/5 |
| MNIST+ 25% | SGD(1.0) | 0/5 | SGD(1.0) | 3/5 | Adam(2e-3, 1e-5) | 3/5 | Adam(1e-3) | 3/5 |
| MNIST+ 50% | SGD(1.0) | 0/5 | SGD(1.0) | 3/5 | Adam(2e-3, 1e-5) | 3/5 | Adam(1e-3) | 3/5 |
| MNIST+ 75% | SGD(1.0) | 0/5 | SGD(1.0) | 3/5 | Adam(2e-3, 1e-5) | 3/5 | Adam(1e-3) | 3/5 |
| MNIST+ 100% | SGD(1.0) | 0/5 | SGD(1.0) | 3/5 | Adam(2e-3, 1e-5) | 3/5 | Adam(1e-3) | 3/5 |
| Vis. Alg. #1 | SGD(1.0) | 0/5 | SGD(1.0) | 3/5 | Adam(2e-3, 1e-5) | 3/5 | Adam(1e-3) | 3/5 |
| Vis. Alg. #2 | SGD(1.0) | 0/5 | SGD(1.0) | 3/5 | Adam(2e-3, 1e-5) | 3/5 | Adam(1e-3) | 3/5 |
| Liar's Puzzle | Adam(2e-3) | 0/15 | Adam(1e-3) | 15/5 | Adam(2e-3, 1e-5) | 15/5 | — | — |
| Vis. Sudoku 10% | SGD(1.0) | 0/100 | SGD(1.0) | 30/15 | Adam(2e-3, 1e-5) | 30/100 | — | — |
| Vis. Sudoku 50% | SGD(1.0) | 0/100 | SGD(1.0) | 30/5 | Adam(2e-3, 1e-5) | 30/100 | — | — |
| Vis. Sudoku 100% | SGD(1.0) | 0/100 | SGD(1.0) | 30/5 | Adam(2e-3, 1e-5) | 0/100 | — | — |

Table 2: Training hyperparameters. Numbers in parentheses after the optimizer denote the learning rate; when there are multiple numbers, different learning rates were applied to different parameter groups, as detailed in the text. Each epoch pair corresponds to the pre-training and training phases, i.e., 3/5 denotes three epochs of pre-training and five subsequent training epochs with the solver layer.

relates the optimizers and epochs for each dataset and solver layer configuration. For SATNet and Scallop, we use the optimization settings described in their respective papers, and found in their public implementations. For SATNet models, the MaxSAT clause parameters were trained at a rate of 2e-3, and the convnet at rate 1e-5. When SGD(1.0) is stated, we used a warmup period spanning the first epoch, and cosine annealing for the remainder of training.

**MNIST Addition.** The MNIST addition problem is the easiest of the problems that we study, at least in its full (100%) configuration. We find that for the 100% configuration, all of the solver layers converge to a nearly optimal solution with three epochs of supervised pre-training, and five epochs of subsequent training with the solver layer attached. For the subsample configurations, all of the solver layers converge to a stable, although in many cases suboptimal, solution within these parameters as well. For SMTLayer, we clipped gradients for all parameters at 0.1, and did not clip gradients for the other solver layer models. For all configurations, we used batches of size 128.

**Visual Algebra.** Although visual algebra is a more difficult learning problem than MNIST addition, as evidenced by the results in Table 1, we find that the same parameters allow all of the configurations studied in our evaluation to converge. After 3/5 epochs of training, the models stabilize, and in some cases, further training yields an overfit model. For SMTLayer, we clipped gradients for all parameters at 0.1, and did not clip gradients for the other solver layer models. For all configurations, we used batches of size 64.

**Liar's Puzzle.** The Liar's Puzzle is the only problem to use a recurrent model, and we found that it required more epochs of pre-training to reduce the variance of the final model with the solver layer. Additionally, the SGD optimizer used with SMTLayer on other datasets caused the model to converge at local minima. We found that pre-training at a higher learning rate, and using Adam with a default learning rate, let to the best results. Additionally, we did not clip gradients for the SMTLayer

**gpt-3.5-turbo and text-davinci-003 System Prompt:**

A Sudoku grid is divided into nine 3x3 subgrids called "boxes". The goal is to fill missing entries, denoted by '_', so that each column, each row, and each box contains all the numbers from 1 to 9, without any repetition.

It is important that numbers given in the puzzle remain unchanged in the solution, and that only numbers from 1 to 9 are used.

Format your solution exactly as shown in the following example, and only provide a solution with no additional commentary.

This is a Sudoku puzzle with each row separated by '|': `{example_instance}`

Here is the solution: `{example_solution}`

Give me a solution to the following puzzle, following the same formatting shown in the example: `{instance}`

Figure 5: Prompt format for OpenAI evaluation.

model. We used the same parameters, but the normal SATNet optimizer, for the SATNet model. For all configurations, we used batches of size 32.

**Visual Sudoku.** Visual Sudoku is the most challenging problem that we studied, for all solver layers as well as the conventional model. We did not use supervised pre-training, as the supervision in this problem leaks the correct labels directly to the model, bypassing the solver layer and the need for its updates [2]. We instead used the unsupervised pre-training method described in [30], and found that 30 epochs of unsupervised pre-training was sufficient to yield consistent and quick convergence with the solver layer. For the most data-scarce configuration (10%), 15 epochs of training with the solver layer were needed to converge, and for the others five epochs were sufficient. For the SMTLayer model, we used batches of size 1 after pretraining (batch size 64 during pre-training), primarily due to the fact that SMTLayer returns only the indices masked as non-hint elements on the Sudoku board. Because each instance has a different number of hint elements, this would lead to ragged tensors during training, which Pytorch does not support.

When assessing SATNet on Visual Sudoku, we were unable to converge to a useful model on any except the full (100%) configuration, as discussed in Section 5.2. Using the authors' public implementation and training script, we attempted the 10% and 50% configurations with and without unsupervised pre-training, with and without the measures taken in [2] to prevent label leakage, and with batches of size 40 (as used in the original paper) as well as 1, to no avail. For the 100% configuration, we were able to reproduce a useful model; we use the accuracy reported in the original paper in Table 1 for consistency, as the average that we obtained did not differ significantly from this.

### A.6 Evaluation of Transformer-Based Approaches at Sudoku Solving

In Section 5, we compared the performance of several large language models with neuro-symbolic techniques in their ability to solve Sudoku puzzles (Table 1). Although the main Sudoku experiments involving SMTLayer were done on the image-based Visual Sudoku benchmark [**?** ], because these models (as well as some of the neuro-symbolic invariants) were only exposed to the rules of Sudoku through training examples, there is a question of whether they are able to learn a representation of the game well enough to perform competitively even when the ground symbols (i.e., hints) are given explicitly. We designed experiments to measure this, interpreting the results as an upper-bound on how well these approaches can perform on Visual Sudoku if they were incorporated with an image feature extractor.

We performed two types of experiments. First, we queried the `gpt-3.5-turbo` and `text-davinci-003` models deployed by OpenAI via their API, which to our knowledge have

not been tuned on a large set of Sudoku puzzles. Second, we fine tuned `gpt-2-xl`, a smaller 1.5 billion-parameter model that was pretrained on WebText for autoregressive language modeling [24]. The prompt used for the experiments is shown in Figure 5. For `gpt-3.5-turbo`, all but the top line was given as the system prompt, and the last was given as the user. For `text-davinci-003`, the entire prompt was given, with the text "Here is the solution:" appended to the end after a newline.

We tuned `gpt-2-xl` on two sets of Sudoku puzzles: the same 9,000 puzzles from the Visual Sudoku training set, and a larger set of 1 million puzzles that we generated. The puzzles in the 1M set were each sampled uniformly from valid Sudoku solutions, and a random subset of 20-75% of the cells in each were masked to create the puzzle.

We fine-tuned GPT2 on four A100 GPUs, using Deepspeed ZeRO stage-2 optimizations [25] and mixed-precision training, for five epochs (74220 steps). The per-device batch size was set to 16 and no gradient accumulation was applied. The Adam optimizer [17] was used ($\beta_1 = 0.9$, $\beta_2 = 0.999$, $\epsilon = $ `1e-8`) with the first 1000 steps for warmup, linear decay, and a maximum learning rate of `5e-5`. Training data was sourced from a specified dataset that contained numerous Sudoku puzzles. The process was conducted over the span of five epochs, with a batch size of 16 set for each of the four GPUs employed. Mixed-precision training was used to increase the speed of training and to save memory without significantly affecting the performance of the model.