# OpenReview forum: "Grounding Neural Inference with Satisfiability Modulo Theories"
_NeurIPS.cc/2023/Conference — NeurIPS 2023 spotlight_

### Official Review · Reviewer_oM4L · 2023-06-23

**Soundness:** 3 good
**Presentation:** 3 good
**Contribution:** 3 good
**Rating:** 7
**Confidence:** 3

**Summary:**

This paper proposes SMTLayer, a layer that incorporates SMT solvers (Z3 in this case) into neural networks. The layer itself is not differentiable. The forward and backward passes of SMTLayer are derived thoroughly. Experiments show that this innovation results in overall more robust, interpretable, and efficient architectures on some tasks where symbolic reasoning is heavily relied on.

**Strengths:**

- The idea itself is straightforward and the derivations seem well-presented.
- The experiments do confirm the claims, and outperform baseline methods by large margins.

### 12 Aug update:
Rebuttal updates and modifications are satisfactory. Bumping up the rating.

**Weaknesses:**

Only some minor weaknesses regarding writing:

- Do not say "[30] showed that ...". If a citation plays a grammatical role, use the authors' names instead.
- ~~Some claims can be made more concrete~~:
  - ~~Line 13: "that are robust to certain types of covariate shift" what types of covariate shift exactly?~~
  - ~~Line 65: "four diverse applications," what applications, how diverse exactly?~~

**Questions:**

NA

**Limitations:**

The paper mentioned a few things as future work, but should do well to comment more on the limitations of the proposed approach.

---

> ### Author Rebuttal · Authors · 2023-08-09
>
> Thanks for you considerable review and we address the reviewer's questions as follows.
>
> > Some claims can be made more concrete. Line 13: "that are robust to certain types of covariate shift" what types of covariate shift exactly?
>
> **A**: In Section 5, we study MNIST addition and Visual Algebra under covariate shift. As discussed in 5.1, we train on 10%, 25%, 50%, and 75% of the possible combinations of digit pairs, and test on 100% of them. For visual algebra, we train on  samples where $a$ and $b$ are the same digit, and $x$ is drawn uniformly from the odd numbers between 0 and 9; we test on samples where $a, b$, and $x$ are drawn uniformly.
>
> In both of these cases, the training and test data are distributed quite differently. Performing well on the test data requires putting the encoded domain knowledge to use effectively.
>
> > Continued: Line 65: "four diverse applications," what applications, how diverse exactly?
>
> **A**: There are four tasks we describe and evaluate our method detailed in Section 5.1, including MNIST Addition, Visual Algebra, Liar’s Puzzle and Visual Sudoku. They are diverse as covering both vision and language tasks, varying from a simple logical problem with fewer variables (e.g. addition) to the more complex one (visual sudoku).
>
> > The paper mentioned a few things as future work, but should do well to comment more on the limitations of the proposed approach.
>
> **A:** Please take the following as a discussion about limitations and will be reflected in writing in camera-ready.
>
> Firstly, a key limitation arises from the necessity for the neural network to interact with the theoretical framework using Boolean vectors. This mandates the discretization of continuous values, potentially resulting in suboptimal inference outcomes. Such discretization can also overcomplicate the theory, leading to further potential performance setbacks.
>
> Secondly, the choice of neural architecture paired with the SMTLayer plays a significant role in determining the quality of representation learned and the final performance, as outlined in Section 5.3. There is a need for more comprehensive research in this area to fully grasp this occurrence and propose potential solutions or modifications.
>
> Lastly, the integration of an SMT solver into a neural network implies a heavier reliance on computations traditionally designated for CPU cores. This can pose a bottleneck, particularly in batched setups. Here, tasks which could be executed concurrently are now being semi-serialized due to the implementation of the proposed method. This not only impacts the speed but also overall computational efficiency. It would be beneficial for the paper to delve deeper into these limitations, offering more insight and potentially suggesting ways to overcome these challenges.
>
> Let us know if we are able to address your concerns (especially on the limitation side of the work). We are happy to respond to any follow-up questions.

---

> > ### Comment · Reviewer_oM4L · 2023-08-11
> >
> > I want to thank the authors for the response and clarify what I meant earlier about making the claims more concrete:
> >
> > For key parts of the paper like abstract and contributions, I think it helps the reader to understand if the texts are self-contained. I understood what the robustness and the diversity refer to after reading the paper, but thought that it might be helpful to have the details explained in the abstract and the contributions. This is why my points are in the "weakness" and not "question" category.
> >
> > The limitation response is satisfactory. I'm willing to adjust my rating if authors can make the writing style modifications.

---

> > > ### Author Response · Authors · 2023-08-11
> > >
> > > Thanks for your fast response to our rebuttal. Yes we see that some ambiguity in the abstract and the contribution may confuse the reader and more details would help to clarify our arguments. We are working to improve the writing and will update the draft when we are given a chance to do so.

---

### Official Review · Reviewer_nvtS · 2023-06-28

**Soundness:** 2 fair
**Presentation:** 2 fair
**Contribution:** 3 good
**Rating:** 6
**Confidence:** 4

**Summary:**

This paper studies neuro-symbolic learning tasks on weakly supervised setting (i.e., lacking direct label supervision of neural networks). To incorporate symbolic knowledge into training, this work integrates SMT solvers into the forward and backward passes of a deep network layer. The key idea is to establish the surrogate gradient of SMT solver-based reasoning, allowing for the back-propagation of this SMT layer. Experimental evaluations on four tasks demonstrate the improvement over several existing approaches.

**Strengths:**

- The paper presents a well-motivated and easy-to-implement method for incorporating symbolic knowledge into network training.
- The formalization derives some good theoretical results, and has proofs presented in the appendix.

**Weaknesses:**

- Theorem 2 requires both the hypothesis set and the loss function to be convex, which is impractical. Moreover, it would be better to decompose this theorem into two parts.
  - The first part is to discuss the convergence of the SGD algorithm. This only requires Lipschitz and smoothness assumptions.
  - The second part is to analyze the properties of the grounding hypothesis under convex assumptions.
- The exactly-one assumption made by Theorem 2 is quite vacuous. From my understanding, with this assumption, one can derive the label supervision by using SMT solver (or correct me if not). Moreover, this assumption avoids the discussion of shortcuts problem (i.e., how to distinguish different satisfying assignments) in such neuro-symbolic learning paradigm [1, 2].
- Some related work is missing. For instance, [3] uses SMT solvers and incorporates MCMC sampling to support network training. Additionally, it is suggested to compare some differentiable logic methods [4, 5]. Particularly, [4] also ensures the satisfaction of symbolic constraints in inference stage.

[1] Zenan Li, Zehua Liu, Yuan Yao, Jingwei Xu, Taolue Chen, Xiaoxing Ma, Jian Lu. Learning with Logical Constraints but without Shortcut Satisfaction.

[2] Marconato, Emanuele, Stefano Teso, and Andrea Passerini. Neuro-Symbolic Reasoning Shortcuts: Mitigation Strategies and their Limitations.

[3] Zenan Li, Yuan Yao, Taolue Chen, Jingwei Xu, Chun Cao, Xiaoxing Ma, Jian Lu. Softened Symbol Grounding for Neuro-symbolic Systems.

[4] Nicholas Hoernle, Rafael Michael Karampatsis, Vaishak Belle, Kobi Gal. MultiplexNet: Towards Fully Satisfied Logical Constraints in Neural Networks.

[5] Zhun Yang, Joohyung Lee, Chiyoun Park. Injecting Logical Constraints into Neural Networks via Straight-Through Estimators.


**Questions:**

- Some citations are duplicate, e.g., [31] and [32], [24] and [25].
- Algorithm 2 is unclear to me. For example, in Line 2, how to compute the BCE loss for $\text{sign}(z)$ in $\\{-1,0,1\\}$. Moreover, the gradient of $\text{sign}(z)$ still vanishes almost everywhere?
- The authors claim that $\ell(\hat{y}, y^*) \leq \ell(y, y^*)$ (Line 211), how to derive this property?
- How does the approach avoid the shortcut problem? For example, in the MNISTAdd task, if the image ``4+7=11`` is incorrectly recognized as ``3+8=11``, the surrogate gradient may reinforce this incorrect prediction instead of correcting it.

**Limitations:**

Yes, the limitations of the approach have been partially presented. The work does not have negative social impacts.

---

> ### Author Rebuttal · Authors · 2023-08-09
>
> Thanks for your considerable and insightful review. We address the reviewer's concerns & questions as follows.
>
> > The exactly-one assumption made by Theorem 2 is quite vacuous. From my understanding, with this assumption, one can derive the label supervision by using SMT solver (or correct me if not). Moreover, this assumption avoids the discussion of shortcuts problem (i.e., how to distinguish different satisfying assignments) in such neuro-symbolic learning paradigm.
>
> **A:** This comment is very insightful and gave us an idea to restructure the proof. Condition (1) just puts some structure on the sample and label sets. Condition (3) is the standard condition needed for convergence of SGD. Condition (2) is needed to establish one-to-one correspondence between labels and constants in Z and needed for using techniques for convergence of SGD, so we can minimize over grounding terms. In other words, assumption (2) allows us to “roughly speaking” move seamlessly between labels and constants in Z. So, if there are some other weaker technical conditions for convergence of SGD, it can easily replace condition (3). Relaxing condition (2) is an interesting open problem. We will clarify this in the revision of the paper. We also give the key ideas in the main paper.
>
> > Some related work is missing. For instance, [3] uses SMT solvers and incorporates MCMC sampling to support network training. Additionally, it is suggested to compare some differentiable logic methods [4, 5]. Particularly, [4] also ensures the satisfaction of symbolic constraints in inference stage.
>
> **A:** Thank you for bringing these to our attention, we will incorporate them into our related work. We do compare the performance of our approach with that of Ahmed et al. (2023), which we see as a method based on differentiable logic. The results are shown in Table 1 (right).
>
> The reason that we chose this as our point of comparison is that the authors used the same set of Sudoku instances to benchmark their work as we consider in our evaluation, and their publicly-available code made it straightforward to reproduce on our hardware. We are happy to include other comparisons, but we are concerned that the code we are able to find for [4] does not have Sudoku constraints, and it is unclear how feasible it is to encode them given the DNF restriction on domain theories in that framework.
>
> >In Algorithm 2, for example, in Line 2, how to compute the BCE loss for $sign(z)$ in {-1, 0, 1}. Moreover, the gradient $sign(z)$ of still vanishes almost everywhere.
>
> **A:** We are taking the gradient w.r.t $z[i]$ instead of $sign(z[i])$ in Algorithm 2 (Line 9), so the gradient does not vanish here. We will clarify this in the writing.
>
> > The authors claim that $\ell(\hat{y}, y^\star) \le \ell(y, y^\star)$, how to derive this property?
>
> This is elaborated further in the proof of theorem 2, which is given in the supplementary material; it is far from obvious that this is where such an explanation would be, and we will provide a pointer in the main text in future drafts.
>
> Let $y^\star$ be an output that achieves smaller loss. Our claim is essentially that the sign of $\hat{y}$ computed on line 3 of both algorithms must be equal to that of $y^\star$ in each dimension. Because the outputs of SMTLayer are always Boolean-valued, this is sufficient, and it follows from two facts:
>
> - At any coordinate $i$ where $y[i] \ne y^\star[i]$, $\mathsf{sign}(\partial_y\ell(y, y^\star))[i] = \mathsf{sign}(y)[i]$.
> - At any coordinate $i$ where $y[i] = y^\star[i]$, $\mathsf{sign}(\partial_y\ell(y, y^\star))[i] = -1 \cdot \mathsf{sign}(y)[i]$.
>
> Because this reasoning doesn’t depend on the specifics of $\ell$, and holds if the loss is not simple cross entropy. This also illustrates why the same reasoning applies if the SMTLayer is embedded deep in the network, as the remainder of the network can be seen as comprising part of $\ell$.
>
> > How does the approach avoid the shortcut problem? For example, in the MNISTAdd task, if the image 4+7=11 is incorrectly recognized as 3+8=11, the surrogate gradient may reinforce this incorrect prediction instead of correcting it.
>
> **A:** This is a great question, and one that we believe requires followup work to investigate. In section 5, we compare the representations learned by SMTLayer versus those learned by Scallop. We found that even when training data is impoverished, the representation learned by SMTLayer is correct with respect to the domain theory on most instances (~99%), whereas Scallop’s is not (50%). We view this as evidence that in these cases our approach successfully avoided the shortcut problem, and our conjecture is that the gradients computed in our backward pass are more sparse, as they are derived from a minimal unsat core. We find that ablating the experiment, replacing minimal unsat core computation with one that instead finds _any_ (non-minimal) set of changes that satisfy constraints, leads to incorrect representations in these cases.
>
> We also point out that our approach does not solve (or claim to solve) this problem in all cases. As we report in the same subsection of section 5, ablating the visual algebra experiments by changing the network’s architecture (monolithic versus segmented) also leads to incorrect representations, and similar end-to-end performance as Scallop. So while SMTLayer seems to improve on past work in this regard, understanding precisely why is important future work.

---

> > ### Comment · Reviewer_nvtS · 2023-08-15
> > **Reply**
> >
> > Thank you for addressing my concerns. I have raised my score to 6.

---

### Official Review · Reviewer_tqWk · 2023-07-04

**Soundness:** 3 good
**Presentation:** 3 good
**Contribution:** 3 good
**Rating:** 6
**Confidence:** 4

**Summary:**

This work proposes to incorporate SMT constraints in the neural network to encode domain knowledge. Specifically, it proposes an unsat core based approach, and an MaxSAT based approach for the differentiable training in the presence of SMT constraints. Empirical evaluations on several benchmark problems are presented.

**Strengths:**

The proposed methods in this work come with theoretical guarantees on convergence as well as significantly better performance than several baseline approaches.

**Weaknesses:**

- The description of the proposed method especially Sec 4 is mostly in English with math hidden behind, which can be confusing for readers who want to know about the technical details. I put some of the technical questions I have in the question part.
- The proposed method seems limited to classification task, without discussions on its generalization.
- For the experiments, only the performance of the unsat core based algorithms is presented and there is not empirical evaluations of the MaxSMT based algorithm presented. The evaluation would be more comprehensive if both qualitative and quantitative comparisons of the two proposed approaches are discussed.

Minor:
- symbol at Line 104 is not defined.
- double quote at Line 251 is not properly typed.

**Questions:**

- At Line 38, it says that the learning of the compatible representation can be done "without requiring label supervision". Still, this work seems to focus on supervised learning. Can the authors elaborate more on what they mean by "without requiring label supervision" to clear the confusion?
- Throughout this work, the loss is always assumed to be binary cross-entropy. I wonder how much this method generalizes to other loss.

More technical questions:
- How are the constraints in the MNIST addition problem defined? In Fig 1, it is unclear why the label is represented by five digits.
- From Sec 4.1, there is a conversion from Boolean vectors to continuous. I wonder how is the number of bits in the vector decided and if it would affect predictive performance.
- How is the amended output \hat{y} computed and is it Boolean or continuous, and how is the inequality at Line 211 is guaranteed?

**Limitations:**

Yes.

---

> ### Author Rebuttal · Authors · 2023-08-09
>
> Thanks for your considerable review. We address the reviewer's questions as follows.
>
>
> > The proposed method seems limited to classification task, without discussions on its generalization. Throughout this work, the loss is always assumed to be binary cross-entropy. I wonder how much this method generalizes to other loss.
>
> **A:** Because the goal of both SAT and MaxSAT is to optimize for the assignment of variables, cross entropy becomes the most natural choice here and in other related work. However, there is no technical limitation that would prevent us from using a type of loss other than cross entropy. This is elaborated further in the proof of theorem 2, which is given in the supplementary material; it is far from obvious that this is where such an explanation would be, and we will provide a pointer in the main text in future drafts.
>
> Let $y^\star$ be an output that achieves smaller loss. Our claim is essentially that the sign of $\hat{y}$ computed on line 3 of both algorithms must be equal to that of $y^\star$ in each dimension. Because the outputs of SMTLayer are always Boolean-valued, this is sufficient, and it follows from two facts:
>
> - At any coordinate $i$ where $y[i] \ne y^\star[i]$, $\mathsf{sign}(\partial_y\ell(y, y^\star))[i] = \mathsf{sign}(y)[i]$.
> - At any coordinate $i$ where $y[i] = y^\star[i]$, $\mathsf{sign}(\partial_y\ell(y, y^\star))[i] = -1 \cdot \mathsf{sign}(y)[i]$.
>
> Because this reasoning doesn’t depend on the specifics of $\ell$, and holds if the loss is not simple cross entropy. This also illustrates why the same reasoning applies if the SMTLayer is embedded deep in the network, as the remainder of the network can be seen as comprising part of $\ell$.
>
> > For the experiments, only the performance of the unsat core based algorithms is presented and there is not empirical evaluations of the **A:** MaxSMT based algorithm presented. The evaluation would be more comprehensive if both qualitative and quantitative comparisons of the two proposed approaches are discussed.
>
> **A:** This is correct. Because MaxSMT requires solving a discrete optimization problem, it is *significantly* more costly than SMT with unsat core tracking enabled. Because we found that the MaxSMT backward pass did not lead to better results on any of our benchmarks, we did not run the full battery of experiments on it. We are more than happy to provide these numbers for the sake of comprehensiveness in future versions of the paper.
>
>
> > At Line 38, it says that the learning of the compatible representation can be done "without requiring label supervision". Still, this work seems to focus on supervised learning. Can the authors elaborate more on what they mean by "without requiring label supervision" to clear the confusion?
>
> **A:** By “without requiring label supervision”, we are referring to the fine-grained labels on an internal representation that would be required to learn, for example, the representations studied in Concept Bottlenecking [Koh et al. 2023]. For example, for learning Visual Sudoku, supervised learning without SMTLayer (or SATNet [Wang et al. 2019]) requires labels of individual digits, rather than just the solution to a set of sudoku puzzles, to learn a dedicated digit classifier first. Our work sidesteps the need to manually break a problem apart in this fashion and obtain intermediate labels to supervise on, and instead allows learning a predictor end-to-end from just labels on the targeted task. We will clarify this when revising the paper.
>
> > How are the constraints in the MNIST addition problem defined? In Fig 1, it is unclear why the label is represented by five digits.
>
> **A:** Figure 1 uses a binary encoding of digits, because SMT solvers support constraints involving integers that are encoded this way. However, only four bits per digit are necessary to represent the inputs, while 5 are needed to represent the output. We attempted to simplify our presentation of this theory by uniformly representing numbers as 5-bit integers, and apologize if this caused confusion. We will make this more clear in the text by discussing the encoding.
>
>
> > From Sec 4.1, there is a conversion from Boolean vectors to continuous. I wonder how is the number of bits in the vector decided and if it would affect predictive performance.
>
> **A:** To make sure we understand this question, first continuous values provided by layers below SMTLayer are converted into a vector of Boolean values by taking their sign. SMTLayer’s output is computed by taking a vector of Boolean values, and making them “continuous” by letting components be -1 for False entries in the vector, and +1 for True entries. The number of bits in each vector is determined by the constraints (domain theory) in the SMTLayer. For visual algebra, for example, there are four digits given as input to the theory, and one digit as output (the solution to the equation). Each digit can be represented in four bits, so the input must be 16 bits, and the output 4 bits.
>
> If one added more bits on either end, it would not impact performance, as the theory would not have a use for them. If one used fewer bits, it would mean that the theory was operating over, e.g., 3-bit integers, which would be unable to represent all of the digits depicted in training and test instances.
>
> > How is the amended output \hat{y} computed and is it Boolean or continuous, and how is the inequality at Line 211 is guaranteed?
>
> **A:** $\hat{y}$ is boolean and constructed at Line 3 in Algorithm 2 (and Algorithm 4), which is to reverse or retrain the sign of each element in $y$ based on the gradient. Please refer to our response to the first question in our rebuttal where we further elaborate how $\hat{y}$ is constructed.
>
> Let us know if we are able to address your concerns and we are happy to respond to any follow-up questions.

---

> > ### Comment · Reviewer_tqWk · 2023-08-16
> >
> > I would like to thank the authors for the clarifications. I'll keep my score.

---

### Official Review · Reviewer_DzNN · 2023-07-05

**Soundness:** 4 excellent
**Presentation:** 4 excellent
**Contribution:** 4 excellent
**Rating:** 8
**Confidence:** 3

**Summary:**


The authors implement SMTLayer, which integrates an SMT solver into a differentiable module, suitable for use with deep learning.  SMTLayer takes a vector of floating-point values as input, and produces a vector of floating-point values as output.  These input vector is  cast to boolean values, and the output of SMTLayer is a set of boolean assignments to variables such that some formula $\phi$ is satisfiable.  The formula is not learned: it represents a constraint that is specified as part of the training task.

In order to calculate a gradient, the problem is run in reverse -- given a gradient on the outputs, SMTLayer will again solve for satisfiability, attempting to find an alternate set of inputs and outputs such that the outputs have lower loss, and $\phi$ is satisfiable.

The authors provide two different SMT solvers for the forward pass, and two different mechanisms for computing gradients on the backwards pass.  They then demonstrate that SMTLayer can be used to solve challenging problems (e.g. MNIST sodoku puzzles) with far greater accuracy than existing SOTA.


**Strengths:**


This paper is very well written, and it addresses an important problem -- namely how to incorporate symbolic and logical computations into deep neural nets in a way that is principled, interpretable, and differentiable.  The experiments are well-designed, the results are compelling, and the authors plan to make the code available as open source.  I believe this to be an important paper with potentially high impact.



**Weaknesses:**


There were a few areas that I did not quite understand -- see questions, below.


**Questions:**


On the backwards pass, SMTLayer must take the gradient of the output $y$, and use it to construct an alternative output $\hat{y}$ with lower loss.  Could you elaborate a bit on how this is done?  What happens if SMTLayer is embedded deep within another neural network, so that you have a gradient on $y$, but that gradient does not come from a simple cross-entropy loss function?  (E.g., there are a bunch of additional NN layers between $y$ and the loss function.)  Do you sample several alternative options for $\hat{y}$, in order to find one that works?  Note that I am not intimately familiar with the details of SMT solvers.

What is typical cost of running an SMT solver, compared to the rest of the neural network?  Does the solver dominate training time?

Can you think of additional areas where an SMT solver could potentially be applied?


**Limitations:**


The authors do not discuss negative societal impacts, but IMO, any impacts are likely to be positive -- incorporating interpretable logical constraints into models seems like an improvement over SOTA wrt. to most issues of AI alignment.

---

> ### Author Rebuttal · Authors · 2023-08-09
>
> Thanks for your considerable review. We address the reviewer's questions as follows.
>
> > Can you elaborate how gradients are computed in the SMTLayer.
>
> **A:** This is elaborated further in the proof of theorem 2, which is given in the supplementary material; it is far from obvious that this is where such an explanation would be, and we will provide a pointer in the main text in future drafts.
>
> Let $y^\star$ be an output that achieves smaller loss. Our claim is essentially that the sign of $\hat{y}$ computed on line 3 of both algorithms must be equal to that of $y^\star$ in each dimension. Because the outputs of SMTLayer are always Boolean-valued, this is sufficient, and it follows from two facts:
>
> - At any coordinate $i$ where $y[i] \ne y^\star[i]$, $\mathsf{sign}(\partial_y\ell(y, y^\star))[i] = \mathsf{sign}(y)[i]$.
> - At any coordinate $i$ where $y[i] = y^\star[i]$, $\mathsf{sign}(\partial_y\ell(y, y^\star))[i] = -1 \cdot \mathsf{sign}(y)[i]$.
>
> Because this reasoning doesn’t depend on the specifics of $\ell$, and holds if the loss is not simple cross entropy. This also illustrates why the same reasoning applies if the SMTLayer is embedded deep in the network, as the remainder of the network can be seen as comprising part of $\ell$.
>
> > What happens if SMTLayer is embedded in another network so gradients on $y$ are not directives from Cross Entropy.
>
> **A:** We use Cross Entropy (CE) as an example in algorithms as we are mostly dealing with assignment problems in this paper, where CE is a natural choice for the loss function and the other related literature. There is no fundamental limitation that would prevent us from using other types of losses and/or more layers between the output of SMTLayer and the groundtruth labels as Theorem 2 (and its proof in the supplementary material) ensures one can construct meaningful gradients over SMTLayers. Please refer to our responses to the question **Can you elaborate how gradients are computed in the SMTLayer. **.
>
>
> > What is typical cost of running an SMT solver, compared to the rest of the neural network? Does the solver dominate training time?
>
> **A:** For the applications studied in Section 5, the solver is typically able to find a solution very quickly. Notice the SMT line in Table 1 (right): Sudoku is the most complex theory that we studied, and it takes 0.05 seconds on average for Z3 (the particular solver we used) to find a solution. The runtime does depend on the complexity of the theory, and there is not a straightforward way to predict how long an SMT solver will take, for example, when there are hundreds or thousands of constraints operating over a similar number of variables. However, we are not at the moment aware of a learning task that would benefit from such complex constraints.
>
> The main performance bottleneck that comes from incorporating an SMT solver on moderately complex theories is parallelism. Solvers run on CPU cores, which will not be as numerous as CUDA cores, so large batches will likely need to be serialized to some extent depending on available resources. Note that our current prototype does not parallelize solver calls, so the performance figures reflected in Table 1 will likely stand to improve from further engineering effort.
>
> > Can you think of additional areas where an SMT solver could potentially be applied?
>
> **A:** Some of the areas where SMT solvers have been useful include program analysis and verification, planning, scheduling, control, model-based design, and systems biology, to name a few. Many of these are also areas that learning has shown good potential, and we are very excited to continue exploring how these tools can be used together to solve interesting problems in all of these domains!
>
> Let us know if the reviewer has more questions or comments and we are happy to respond.

---

> > ### Comment · Reviewer_DzNN · 2023-08-14
> > **Further questions...**
> >
> >
> > I'm not quite sure you answered my question...
> >
> > > Let $y^{\star}$ be an output that achieves smaller loss...
> >
> > Okay, but how to you get $y^{\star}$?  If the SMTLayer feeds directly into a cross-entropy loss function, then you have ground truth for what $y$ --- the output of the SMTLayer --- is supposed to be, and thus can trivially find a $y^{\star}$ with lower loss.  However, if the SMTLayer is embedded deep within another neural network with some arbitrary loss function (i.e. there are many NN layers between $y$ and the loss), then all you have is $y$ and a gradient on $y$.  Since $y$ is boolean-valued, simply applying the gradient with a small learning rate (as is usual for gradient descent) will yield a non-boolean $y^{\star}$ that is very close to the original $y$, and unlikely to differ in sign.  So you probably need to sample different possible boolean outputs, perhaps informed by the gradient, and re-run the latter half of the NN to see if the loss goes down, which is highly non-trivial.  Am I missing something here?
> >
> > It's perfectly fine to say: "in our implementation the SMTLayer must be connected directly to the loss function, and only some losses are supported," I just want to be clear about it.

---

> > > ### Author Response · Authors · 2023-08-15
> > > **Clarification**
> > >
> > > > Since is boolean-valued, simply applying the gradient with a small learning rate (as is usual for gradient descent) will yield a non-boolean that is very close to the original, and unlikely to differ in sign
> > >
> > > That is absolutely correct. We compute $\hat{y}$ via $\mathsf{sign}(y) - 2 \times \mathsf{sign}(\partial_y)$, rather than the usual gradient descent update. Because $y$ and $\hat{y}$ are boolean-valued, we only care about the sign of the gradient, and not its magnitude.
> > >
> > > We will clarify the writing around this point, as it is subtle. We also want to be clear that we have only done experiments that use cross-entropy loss, with the SMT layer connected at the top. While we believe that the same underlying reasoning applies to other losses and configurations of the layer within a network, we do not claim to show that this is effective in practice on any benchmarks.

---

### Decision · Program_Chairs · 2023-09-21

**Decision:**

Accept (spotlight)

**Comment:**

The submission present SMTLayer, a method to integrate a SMT solver as a differentiable module into a (deep) neural network. It allows model designers to specify logical constraints as a fixed formula input $\varphi$ to the SMTLayer, which during training/inference interprets (floating point) inputs generated by prior layers as boolean values and outputs boolean values corresponding to a satisfying model of $\varphi$.

All reviewers agree that this is a novel and interesting combination of symbolic and neural methods. In particular, the fact that the paper has both theoretical grounding as well as convincing empirical results is highlighted.

Weaknesses raised by the reviewers were mainly related to the presentation or misunderstandings, which the authors could clear up during the rebuttal phase.

Overall, this is a strong paper that deserves to be presented at NeurIPS.